# Evolutionarily informed machine learning enhances the power of predictive gene-to-phenotype relationships

Chia-Yi Cheng[1,5], Ying Li[2,3], Kranthi Varala [2,3], Jessica Bubert [4], Ji Huang [1], Grace J. Kim [1], Justin Halim[1], Jennifer Arp[4], Hung-Jui S. Shih[1], Grace Levinson[1], Seo Hyun Park [1], Ha Young Cho[1], Stephen P. Moose [4] & Gloria M. Coruzzi [1✉]

Inferring phenotypic outcomes from genomic features is both a promise and challenge for systems biology. Using gene expression data to predict phenotypic outcomes, and functionally validating the genes with predictive powers are two challenges we address in this study. We applied an evolutionarily informed machine learning approach to predict phenotypes based on transcriptome responses shared both within and across species. Specifically, we exploited the phenotypic diversity in nitrogen use efficiency and evolutionarily conserved transcriptome responses to nitrogen treatments across Arabidopsis accessions and maize varieties. We demonstrate that using evolutionarily conserved nitrogen responsive genes is a biologically principled approach to reduce the feature dimensionality in machine learning that ultimately improved the predictive power of our gene-to-trait models. Further, we functionally validated seven candidate transcription factors with predictive power for NUE outcomes in Arabidopsis and one in maize. Moreover, application of our evolutionarily informed pipeline to other species including rice and mice models underscores its potential to uncover genes affecting any physiological or clinical traits of interest across biology, agriculture, or medicine.

[1] Department of Biology, Center for Genomics and Systems Biology, New York University, New York, NY 10003, USA. [2] Department of Horticulture and Landscape Architecture, Purdue University, West Lafayette, IN, USA. [3] Purdue Center for Plant Biology, Purdue University, West Lafayette, IN, USA. [4] Department of Crop Sciences, University of Illinois at Urbana-Champaign, Urbana, IL 61801, USA. [5] Present address: Department of Life Science, National Taiwan University, Taipei, Taiwan. ✉email: gloria.coruzzi@nyu.edu

Being able exploit genomic data to predict organismal outcomes in response to changes in nutrition, toxin, and pathogen exposure could inform crop improvement, disease prognosis, epidemiology, and public health. To this end, machine learning methods have been developed and applied to infer phenotypes from genomic and epigenetic features associated with such conditions using changes in mRNA/protein expression levels, single nucleotide polymorphisms, chromatin modifications, and more. Despite the compelling motivation and cumulative efforts, accurately predicting complex phenotypic traits from genome-scale information remains both a promise and a challenge.

Several factors contribute to these challenges. First, in contrast to the increasing availability of omics data, collection of high-quality phenotypic data from a genetically diverse population that adequately represents the phenotypic diversity space has become a major limiting factor[1]. In addition, phenotypic data is often collected from experiments that are distinct from those used to acquire the functional genomics data. To overcome these limitations, phenotyping efforts should be expanded and performed on the same materials that are the source of genetic/genomic information[2]. Furthermore, the explosion of omics data means that the features (e.g., numbers of genes) collected from a single experiment inevitably outnumber the phenotype space (e.g., sample size), leading to problems in data sparsity, multi-collinearity, multiple testing, and overfitting[3]. This can be counteracted with increasing sample size, dimension reduction, or feature selection methods such as Principal Component Analysis (PCA), Least Absolute Shrinkage and Selection Operator (LASSO) regularization, Canonical Correlation Analysis (CCA), and so forth[4]. Additionally, cross-species approaches have been adopted in the machine learning context to improve the performance of model-to-human knowledge translation[5].

Herein, we address a number of these challenges by using an evolutionarily informed machine learning approach that exploits genetic diversity both within and across species. In a proof-of-principle study with practical implications, we employ transcriptome data of nitrogen response genes to predict nitrogen use efficiency (NUE), an agronomic outcome critical for worldwide food safety and sustainability[2,6]. Nitrogen (N)—the main limiting macronutrient for plant growth—is supplemented in agricultural systems through application of N fertilizer. For major row crops such as maize (*Zea mays*), less than 40% of supplied N is taken up by the plants, while more than 60% of soil N is lost to the atmosphere or water bodies through multiple processes such as denitrification, ammonia volatilization, leaching etc[7]. Balancing the need to further increase crop yields, while also mitigating the environmental impacts associated with N fertilizer, is a challenge for sustainable agriculture. Considering the polygenic nature of NUE that involves the integration of developmental, physiological, and metabolic processes[2], machine learning is an appealing strategy to tackle the mechanisms underlying this complex trait.

To this end, we collected transcriptomic and phenotypic NUE data from two species—maize (a crop) and Arabidopsis (a model)—each of which included a panel of genotypes with diverse genetic background and NUE variation. We used genes, whose response to N-treatments (N-DEGs) was conserved within and across species as a dimension reduction approach for machine learning. As maize and Arabidopsis are highly divergent phylogenetically, these evolutionarily conserved N-response genes should represent essential/core functions contributing to NUE. We show that models constructed using these evolutionarily conserved N-DEGs significantly improved the prediction of NUE traits from gene expression values, compared to an equal number of top ranked N-DEGs or randomly selected expressed genes. Importantly, the inclusion of the model species Arabidopsis in our study enabled us to validate our findings using mutants. This experimental evidence validated that the genes, whose expression levels are important in predicting NUE in the machine learning models are more than just markers, but functionally required for the trait. Moreover, we show that our evolutionarily informed machine learning pipeline is transferable to other species and traits in plants and animals. Specifically, application of our method to other matched transcriptome and phenotype datasets related to drought in field grown rice or disease in mouse models resulted in enhanced prediction accuracies of the learned models. As such, our evolutionarily informed machine learning pipeline has the potential to identify genes of importance for complex phenotypes of interest across biology, agriculture, or medicine.

## Results

**Overview: evolutionarily informed machine learning pipeline enhances the predictive power of a gene expression-to-trait analysis.** The goal of this work was to test whether the prediction power of machine learning models could be enhanced by exploiting the genetic diversity of gene responses and phenotypes both within and across species. To this end, in our proof-of-principle study, we tested whether using N-responsive differentially expressed genes (N-DEGs) conserved both within and across species as a biologically-principled means of dimension reduction, could enhance our ability to learn genes of importance to predicting NUE phenotypes from gene expression data across a model (Arabidopsis) and crop (maize) plant. This model-to-crop machine learning approach also allowed us to more rapidly validate conserved features of importance to NUE in the crop using the model species.

*Genetic diversity for NUE phenotype.* Within each species, we selected a set of genotypes that exhibit a broad spectrum of phenotypic variation in NUE. Our data included 18 Arabidopsis accessions that were previously identified for their NUE diversity[8], which originated from a nested collection of 265 natural accessions found in a wide range of habitats differing notably in soil nutrient richness[9]. The 23 maize genotypes used in our study, correspond to 12 maize inbred lines and their 11 corresponding hybrids with B73. We selected these 12 maize inbred lines to represent the phenotypic diversity for NUE traits that we measured among a population of 318 field-grown maize inbreds (Supplementary Data 1 and Supplementary Fig. 1), which broadly represent the current germplasm base for U.S. Corn Belt hybrids. This maize population that we tested for NUE traits includes the parents of the Nested Association Mapping (NAM) population[1], improved inbreds from different breeding programs described in recently expired plant variety patents[10], and the Illinois Protein Strains that display the known phenotypic extremes for NUE traits in maize[11,12] (Supplementary Data 1). The B73 inbred maize line was chosen as the parent for the hybrids, because it is a major founder of the Stiff-Stalk heterotic group used in the production of nearly all commercial U.S. Corn Belt hybrids[13]. Furthermore, B73 displays high nitrogen utilization efficiency (NUE) (Supplementary Data 1), and also serves as the reference genome sequence assembly for maize[14].

Next, to test whether genome-wide responses to N-treatments evolutionarily conserved across the model and crop could be a biologically principled approach to enhance the model performance of predicting NUE, we constructed a three-step machine learning pipeline (Fig. 1).

**Step 1 feature selection.** We collected and analyzed matched phenotypic and transcriptomic data from the same replicate plants for each N-treatment conducted in a controlled laboratory setting (Arabidopsis) or field conditions (maize) and

## Evolutionarily informed machine learning pipeline

**Fig. 1 Evolutionarily informed machine learning approach enhances the predictive power of gene-to-phenotype relationships.** Step 1 feature selection: Phenotypic and transcriptomic data of N-responses were generated from Arabidopsis (lab-grown) and maize (field-grown) under low-N vs. high-N conditions. The expression levels of N-response differentially expressed genes (N-DEGs) conserved in both species were identified via "leave-out-one" approach (Fig. 4) and used as gene features in the machine learning methods in Step 2. This biologically principled approach to reduce the feature dimensions ultimately improved the model performance (Table 1). Step 2 feature importance: We ranked the genes based on (i) the XGBoost-derived feature importance score (left) and (ii) the TF connectivity in a GENIE3 regulatory network (right) constructed from the N-response TFs (Step 1) as regulators and the XGBoost important features as targets. Step 3 feature validation: We validated the role of NUE for eight TFs in planta using Arabidopsis and maize loss-of-function mutants.

(Supplementary Fig. 2). Using linear models, we identified N-response differentially expressed genes (N-DEGs) in parallel for maize and Arabidopsis, and retained the N-DEGs conserved both within and across species as gene features used in machine learning. **Step 2 feature importance.** We selectively used the expression levels of these evolutionarily conserved N-DEGs, as a biologically-principled approach to feature reduction in the gradient boosting-based method XGBoost[15] predictive models. The outcome of the machine learning enabled us to rank the N-DEGs whose expression levels best predicted the NUE traits measured in the same set of plants. Moreover, we inferred the transcription factors (TF) regulating these genes of importance to NUE and measured their connectivity in the NUE network by constructing a NUE gene regulatory network (GRN) using a Random Forest-based method GENIE3[16]. Through integration of

the results of these complementary means, we generated ranked lists of: (i) gene features based on their contribution to the trait prediction (XGBoost-based importance score), and (ii) TFs based on their level of connectivity in the GRN for each species (GENIE3-based connectivity). **Step 3 feature validation.** We validated the function of eight candidate TFs in Arabidopsis or maize based on their importance score to the NUE trait and/or their degree of connectivity in the GRN. We experimentally confirmed the function of these eight TFs in regulation of NUE in planta using loss-of-function mutants in Arabidopsis, as well as in maize, where available. Detailed descriptions of our evolutionarily informed cross-species machine learning analysis pipeline can be found in "Methods" section.

**Quantifying NUE phenotypes across Arabidopsis and maize varieties.** In our phenotypic analysis, we quantified nitrogen use efficiency (NUE) as the efficiency of converting supplied N to biomass/grain yield. For Arabidopsis, NUE was calculated as the efficiency with which each plant converted supplied N into shoot biomass (NUE = Above ground dry weight/Applied N). This measure of NUE is achieved by providing each plant with a trackable/contained amount of N in pots in a lab setting, as a proxy for the field agricultural setting[2]. Indeed, we found the Arabidopsis accessions previously selected for NUE diversity[8] present a broad range of NUE variation in our own experiments, as evidenced by the coefficient of variation (CV = 0.58) (Fig. 2a). The correlation of traits shows that NUE at the pre-bolting stage is highly correlated with NUpE ($r = 0.88$), and to a lesser extent with NUtE ($r = 0.39$) (Fig. 2b). The NUE variation among the Arabidopsis accessions is primarily explained by nitrogen levels, followed by accession and nitrogen-by-accession interaction (Two-way ANOVA $P$-value: **G**, <2E−16; **N**, <2E−16; **G × N**, 9.93E−07). This indicates the N-level explains the phenotypic variation in NUE in this collection of Arabidopsis ecotypes.

For field-grown maize, we used Total NUtE, (stover biomass + grain biomass)/(stover N content + grain N content), as the target trait (Fig. 3a). We chose this because Total NUtE is more robust to the effects of maturity and photoperiod in the field[17] (Supplementary Fig. 3), and remains highly correlated to grain NUtE (Fig. 3b). We measured total NUtE across 318 maize inbred lines in a field experiment where soil N supply was not limiting, and observed a nearly three-fold range in total NUtE (56–156 kg biomass/g plant N) (Supplementary Data 1 and Supplementary Fig. 1). To illustrate the influence of soil N-supply on total NUtE, in a pilot study, 25 inbred maize lines chosen to represent both historical (NAM parents)[1] and elite genetic diversity[10] were grown in adjacent plots that received either no N fertilizer or were N-fertilized as the larger population (see "Methods" section). When grown with sufficient N, the distribution of NUtE values for these 25 maize inbreds overlaps with that observed from the larger population of 318 maize genotypes (Supplementary Fig. 2). For this study, we selected 12 (from the 25 above) maize inbreds, which exhibited a similar coefficient of variation for NUtE phenotypic values (CV = 0.19) as the larger population of 318 genotypes (CV = 0.15) for matched transcriptome profiling and detailed phenotyping in N-responsive field plots, over three field seasons (Supplementary Data 1).

ANOVA results revealed that 55% of the total NUtE variation in this maize experiment was attributed to genetic effects (Fig. 3c). Our two-way ANOVA analysis of the maize data shows that in addition to $G$ ($P$-value = 8.6E−11) and $N$ ($P$-value = 2.9E−13), $G × N$ was also a significant factor ($P$-value = 2.28E−07) explaining 19% of the variation in Total NUtE (Fig. 3c). This is distinct from our findings for Arabidopsis, where $N$ is the main explanatory variable (Fig. 2c). This difference likely reflects not

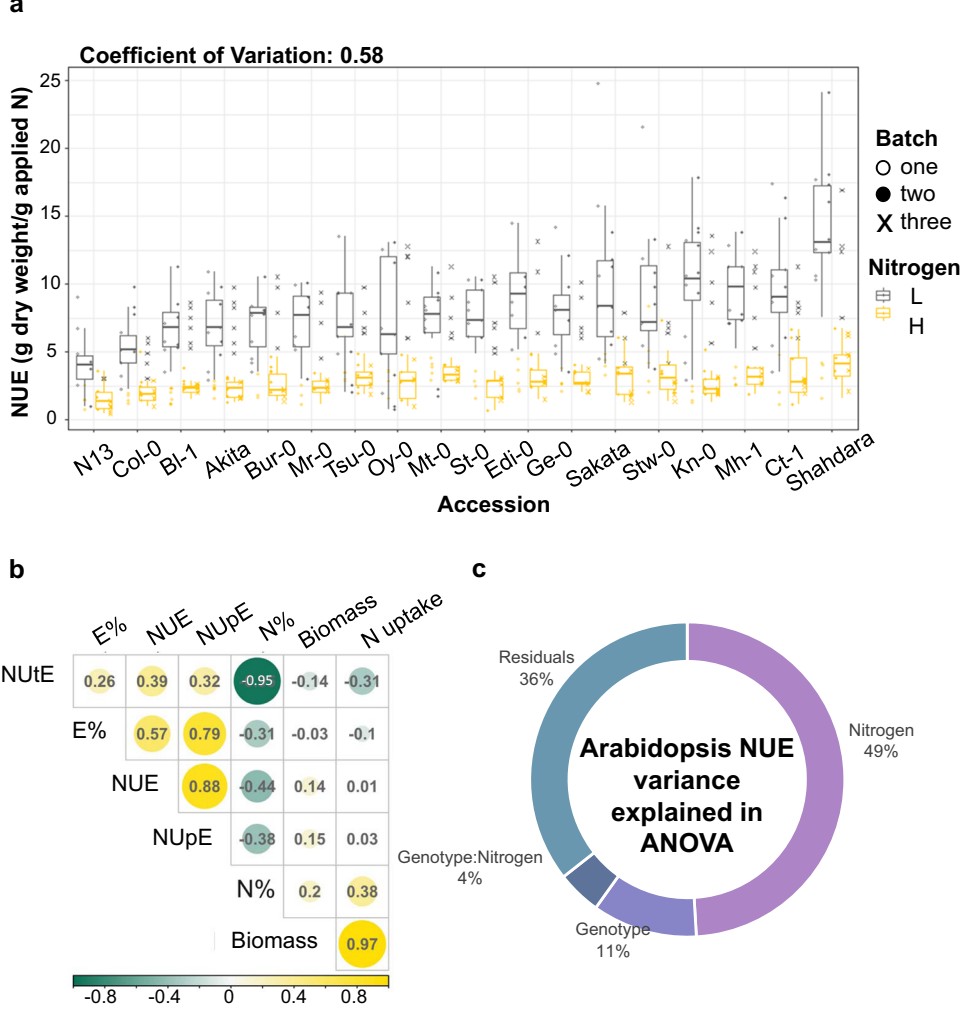

**Fig. 2 Nitrogen is the leading factor explaining the NUE variation across Arabidopsis natural acesssions. a Boxplot of NUE among the Arabidopsis genotypes measured in three independent batches.** The coefficients of variation demonstrate the broad range of phenotype of this panel of genotypes, which has been widely used in NUE studies. The X-axis is ordered in the increasing value of average NUE. In the box plots, the box represents the 25th to 75th percentile and the line within the box marks the median. Whiskers above and below the box indicate the 10th and 90th percentiles. Points above and below the whisers indicate outliers outside 10th and 90th percentiles. **b The correlation of traits measured in this study**. NUE at the pre-bolting stage is highly correlated with NUpE. Biomass, g/plant; N uptake, mg N/plant; N%, N uptake/Biomass; E%, 15N uptake/N uptake; NUE, Biomass/applied N; NUpE, 15N uptake/applied 15N; NUtE, Biomass/N uptake. **c The NUE variation is primarily explained by nitrogen levels, followed by accession and nitrogen by accession interaction.** Two-way ANOVA P-value: $G$, $< 2E{-}16$; $N$, $< 2E{-}16$; $G \times N$, 9.93E$-$07. For each genotype $n > 10$ biologically independent plants examined over three independent experiments. The source data for this figure is provided in Supplementary Data 1.

only the overall greater genetic diversity in the maize varieties, but also suggests that intensive breeding and selection for N-responsive grain yields in maize[18] may have expanded the phenotypic variation for NUE beyond that observed among the Arabidopsis natural accessions. We therefore included these important interactions of maize genotype with nitrogen supply on the NUE phenotype as a factor in our computational pipeline described below.

**Evolutionarily conserved transcriptome response to N-treatment used for feature reduction in machine learning**. Feature reduction is an essential pre-processing step in machine learning, as too many irrelevant features may interfere with prediction performance[3]. Given the fact that the N level is a significant factor explaining NUE variation in both Arabidopsis and maize (Figs. 2c and 3c), we used negative binomial Generalized Linear Mixed models (GLMs) in edgeR R-package[19] and identified N-DEGs

(Gene expression ~ Condition + Genotype) in the training data ($n$ $-1$ genotype). Importantly, we note that the testing data sets (the held-out genotype) were never used to select the N-DEGs. This was repeated in a round-robin manner across genotypes for each species (Supplementary Fig. 4). Next, we retained the evolutionarily conserved N-DEGs by mapping the Arabidopsis N-DEGs to their corresponding maize homologs using Phytozome 10 [20] (Fig. 4 and Supplementary Data 8, Cross-species Feature Reduction). This cross-species analysis enabled us to (i) apply an evolutionarily guided filter to reduce the dimensionality of gene features used in machine learning, and (ii) enhance our ability to perform rapid validation testing of candidate NUE genes with relevance to the crop in the model species.

The resulting conserved N-DEGs from Arabidopsis ($n = 610$) (Supplementary Data 3) were used as gene features in the machine learning model (Fig. 5). We further subjected the conserved N-DEGs from maize to a second round of filtering to identify those also responding to $N \times G$ interaction (Fig. 4,

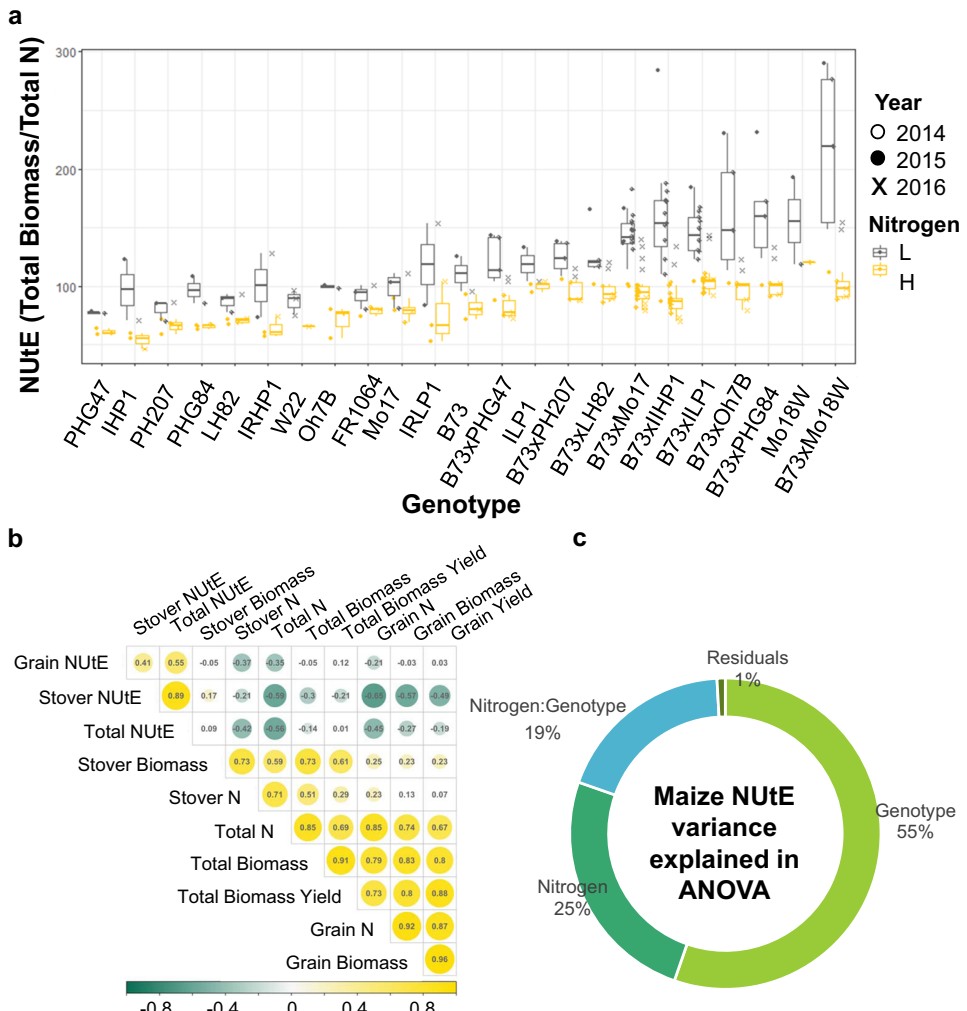

**Fig. 3 Genotype is the leading factor explaining the NUE variation in maize breeding lines. a Boxplot of total nitrogen utilization (NUtE) values among the maize genotype panel measured in three consecutive years.** The *X*-axis is ordered by increasing value of average total NUtE. The coefficients of variation demonstrate the broad range of phenotype of this smaller panel of maize genotypes, which spans the distribution of NutE values measured in a larger representative germplasm collection (Supplementary Fig. 2). In the box plots, the box represents the 25th to 75th percentile and the line within the box marks the median. Whiskers above and below the box indicate the 10th and 90th percentiles. Points above and below the whisers indicate outliers outside 10th and 90th percentiles. **b The correlation of traits measured in this study.c** The total NUtE variance of 2014, the year when the RNA samples were harvested, is primarily explained by Genotype (*G*), followed by *N*, and *G* × *N* effect. Two-way ANOVA *P*-value: *G*, 8.6E−11; *N*, 2.9E−13; *G* × *N*, 2.28E−07. For each genotype *n* > 5 biologically independent plants examined over three independent experiments. The source data for this figure is provided in Supplementary Data 1.

Within-species Feature Reduction). This second filter aimed to account for the significant $N \times G$ effect that we observed in the maize NUE phenotypes (Fig. 3c), resulted in a list of maize N-DEGs responsive to $N \times G$ interaction ($n = 248$) (Supplementary Data 3). Next, these two sets of conserved N-DEGs from Arabidopsis and maize were used as features in the machine learning model (Fig. 5).

We then tested the hypothesis that the expression levels of N-DEGs conserved across model and crop species could enhance our ability to infer NUE phenotypes—compared to non-selected genes—using machine learning algorithms. This data-driven hypothesis is supported by the fact that: (i) the expression levels of N-DEGs have been used as biomarkers of N status across maize genotypes[21], and (ii) our phenotypic data shows that N level is a significant factor explaining the NUE variation in both maize and Arabidopsis (Figs. 2c and 3c). Indeed, this analysis enabled us to determine that the predictive performance of our models learned is significantly better at predicting NUE outcomes when the

evolutionarily conserved N-DEGs are used, compared to the same number of top-ranked N-DEGs with the lowest *P*-value, or randomly selected expressed genes (Table 1), as detailed below.

**Evolutionarily conserved N-responsive genes have enhanced predictive power in machine learning.** For each species, we used the gene expression values (N-DEGs) as features (gene features hereafter) to predict NUE traits through XGBoost regression models. XGBoost[15] is a implementation of the gradient boosting algorithm[22], that uses a boosting algorithm to combine multiple weak learners, i.e. shallow trees, into a strong one (Fig. 5, Step 2). Lastly, we used the trained XGBoost models to predict NUE for the left-out genotype and evaluated the model performance using correlation between the observed- and the predicted-NUE in the left-out test set (Fig. 5, Step 3). In summary, we repeated the above steps and constructed 18 models for Arabidopsis, and 16 models for maize, corresponding to each genotype analyzed (See Supplementary Fig. 4 for a detailed illustration).

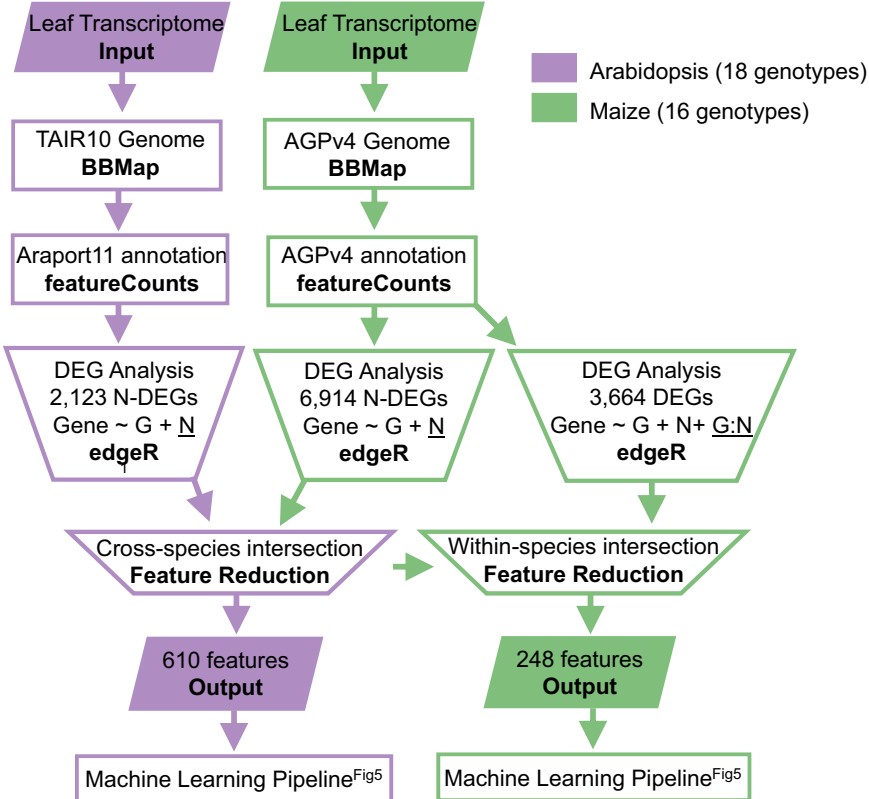

**Fig. 4 Evolutionarily conserved N-response genes across Arabidopsis-maize used as a biologically principled feature reduction method for the XGBoost machine learning pipeline.** The RNA-seq reads from leaves of Arabidopsis and maize N-treated samples were aligned to reference genome assemblies using BBMap and the read counts were generated using featureCounts. The N-response DEGs (N-DEGs) were identified using generalized linear models in edgeR and leave-out-one method: one genotype (out of 18) was left out during each round of analysis and the intersection of 18 DEG lists was used for feature reduction (For details, see Fig. S4). The overlap of N-DEGs from Arabidopsis ($n = 2123$) with maize ($n = 6914$) resulted in a set of evolutionarily conserved N-response Arabidopsis genes ($n = 610$), which were used as features in the machine learning model. The corresponding conserved N-response genes in maize were further intersected with genes responding to nitrogen by genotype effects ($n = 3664$), resulting in 248 maize genes that were used as features in the machine learning model to predict NUE. The complete ranking of DEGs is provided in Supplementary Data 2.

For maize, using the N-DEGs ($n = 248$) conserved with their Arabidopsis homologs, resulted in a mean Pearson's correlation coefficient $r$ of 0.79 for the XGBoost models predicting NUE across 16 maize lines (Fig. 5, Step 3). The $r$ was above 0.6 for all but two maize genotypes, Illinois High Protein (IHP1) and Illinois Low Protein (ILP1). These two maize inbred line are derived from more than 100 cycles of divergent selection for seed protein concentration and other component traits of nitrogen use efficiency[11,12]. Thus, it is not surprising that the models showed lower accuracy in predicting the NUE phenotypes of IHP1 and ILP1, compared to other maize inbreds and the hybrids that each share the B73 parent.

Importantly, our analysis showed that the overall predictive performance of learned models that used the evolutionarily conserved maize N-DEGs is significantly better than that obtained using the same number of top-ranked N-DEGs with the lowest P-value (0.68, Mann–Whitney U-test P-value = 1.06E−3), or ones randomly selected from total expressed genes (0.62, Mann–Whitney U-test, P-value = 1.5E−5) (Table 1). In addition, our comparison of the feature importance score, an XGBoost[15] output which reveals the influence of each feature (gene) in the predicted value (NUE)[15], with the P-value in DEG analysis, uncovered only a weak correlation (Spearman's rank correlation coefficient $\rho = 0.19$, Supplementary Fig. 5b). These comparisons support the interpretation that XGBoost models capture non-linear gene–trait relationships and our hypothesis that evolutionarily conserved N-DEGs enhance the machine learning outcome.

In parallel, we used the Arabidopsis N-DEGs ($n = 610$) whose N-response is conserved with their maize homologs, as the features to predict NUE in the same XGBoost machine learning pipeline (Fig. 5). Our machine learning results show that the mean Pearson's correlation coefficient $r$ across all 18 Arabidopsis genotypes was 0.65 (Fig. 5, Step 3). Moreover, we found that this overall model performance is significantly better than that obtained using the same number of top-ranked N-DEGs with the lowest P-value ($r = 0.59$, Mann–Whitney U-test P-value = 1.64E−4), or ones randomly selected from total expressed genes ($r = 0.53$, Mann–Whitney U-test, P-value = 3.82E−6) (Table 1). Similarly, we found that the feature importance ranking was weakly correlated with the edgeR-based P-value ranking of DEGs (Spearman's rank correlation coefficient $\rho = 0.14$, Supplementary Fig. 5a).

Overall, our results from both maize and Arabidopsis data show that using the evolutionarily conserved N-responsive differentially expressed genes significantly improved performance of the machine learning models predicting NUE and that this improvement is not due to a simple numerical reduction in the gene features (Table 1). Furthermore, the weak correlation between the XGBoost-based feature importance ranking and the edgeR-based P-value ranking (Supplementary Fig. 5), indicates that XGBoost can capture non-linear gene–trait relationship beyond single variable DEG analysis. It is also worth emphasizing that we used one set of hyperparameters for each species to achieve a consistent performance across genotypes, suggesting that the model is generalized and likely applicable to additional genotypes. Taken together, our results

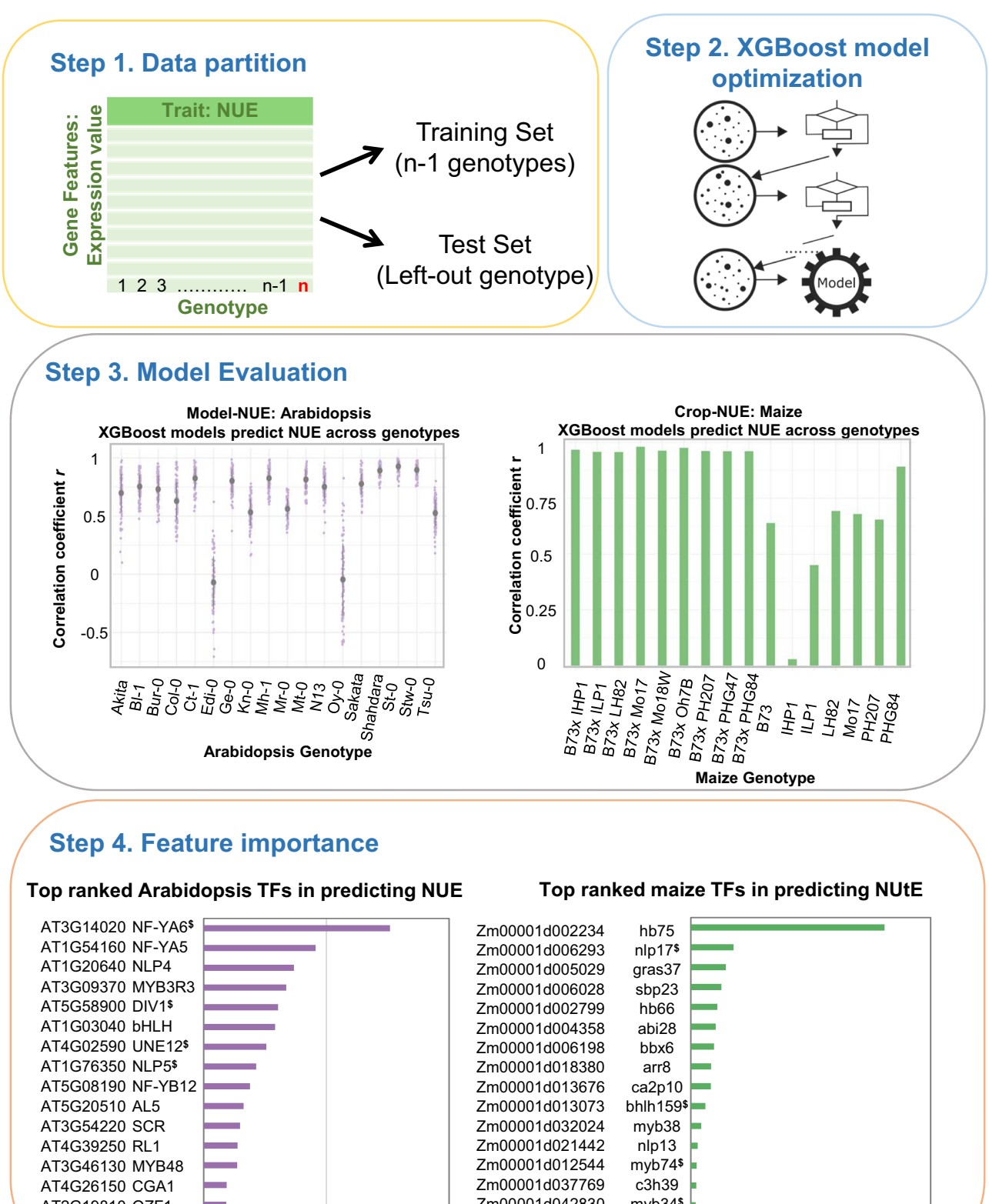

demonstrate that NUE—a polygenic trait—could be predicted from gene expression levels of N-DEGs, and that using an evolutionarily principled approach to feature reduction significantly improved the model performance.

**Predicting additional traits demonstrates the general applicability of the evolutionarily informed machine learning pipeline.** To further test whether our pipeline can be applied to predict additional traits from transcriptome data, we used the

**Fig. 5 Evolutionarily informed machine learning models uncover genes-of-importance and predictive of NUE. Step 1**. The evolutionarily conserved N-DEGs between Arabidopsis and maize (see Fig. 4) and NUE data from $n$ genotypes are split into training ($n-1$ genotypes) and test (left-out genotype) set (for details see Supplementary Fig. 4). **Step 2**. The training set was used to optimize the XGBoost model, which then predicts the NUE using the gene expression in the test set. **Step 3**. The model performance was evaluated by calculating the Pearson's correlation coefficient $r$ between the predicted and actual NUE values. In Arabidopsis, the dots indicate the Pearson's $r$ of 100 individual iterations and the pointranges indicate mean $+/-$ SD. In maize, there are only two data points for each genotype thus the Pearson's $r$ was calculated from the pooled predicted and actual NUE from 100 iteration. **Step 4**. The TF features were ranked based on their contribution to the NUE. The complete ranking of gene features is provided in Supplementary Data 3. $^\$$These genes are functionally validated in this study.

**Table 1 Evolutionary conservation of gene responsiveness enhances machine learning outcomes.**

| Maize features | Random expressed genes | Top N-DEGs | Cross species N-DEGs |
|---|---|---|---|
| **Pearson's $r$** | $r = 0.62$ | $r = 0.68$ | $r = 0.79$ |
| Random expressed genes | | 6.56e−04 | 1.5e−05 |
| Top N-DEGs | 6.56e−04 | | 1.06E−03 |
| Cross species N-DEGs | 1.5e−05 | 1.06−03 | |

| Arabidopsis features | Random expressed genes | Top N-DEGs | Cross species N-DEGs |
|---|---|---|---|
| **Pearson's $r$** | $r = 0.53$ | $r = 0.59$ | $r = 0.65$ |
| Random expressed genes | | 7.63E−06 | 3.82E−06 |
| Top N-DEGs | 7.63E−06 | | 1.64E−04 |
| Cross species N-DEGs | 3.82E−06 | 1.64E−04 | |

Comparison of the performance of maize (top) and Arabidopsis (bottom) XGBoost models using the same number of features from different sources: randomly selected expressed genes, top N-DEGs based on FDR ranking in edgeR analysis, and the evolutionarily conserved N-DEGs. The numbers indicate the $P$-value of one-tailed Mann–Whitney $U$-test.

same conserved N-DEGs (Fig. 4), to predict two additional traits for each species. For Arabidopsis, we found that the mean Pearson's $r$ for predicting biomass and N-uptake was 0.68 and 0.69, respectively (Supplementary Fig. 6a), is comparable to that for predicting NUE ($r = 0.65$). Interestingly, the feature importance ranking appeared to be trait-specific, as the gene ranking for NUE only weakly correlated with those for biomass ($\rho = 0.09$) and N-uptake ($\rho = 0.08$) (Supplementary Figs. 6c and 7b). This result can be explained by the weak correlation between NUE and biomass ($r = 0.14$), as well as that between NUE and N-uptake ($r = 0.01$) (Fig. 2b). Indeed, for highly correlated traits such as biomass and N-uptake ($r = 0.97$), the feature importance rankings were also highly correlated ($\rho = 0.94$) (Supplementary Fig. 7a). For maize, the mean Pearson's $r$ for predicting biomass and grain yield was 0.72 and 0.52, respectively (Supplementary Fig. 6b). As with Arabidopsis, the feature importance rankings for maize also appeared to be trait-specific, being greater ($\rho = 0.59$) for highly correlated traits such as biomass and grain yield ($r = 0.8$), compared to Total NUtE—which is weakly correlated with either biomass ($r = -0.14$; $\rho = 0.15$) or grain yield ($r = -0.19$; $\rho = 0.33$) (Fig. 3b and Supplementary Fig. 8). Taken together, our results suggest that the feature importance ranking can capture biological information represented by the degree of phenotypic correlation among different component traits.

To extend our studies beyond our proof-of-principle dataset, we also applied our evolutionarily informed machine learning pipeline to two additional matched transcriptome and phenotype datasets related to drought in field grown rice and disease response in mouse models.

The rice data comprises matched transcriptomic and phenotypic information collected from 220 rice genotypes subjected to drought treatment in field experiments[23]. The 220 rice genotypes consist of two major subspecies, Indica and Japonica, which diverged ~440,000 years ago, with the genotypic and phenotypic diversity of domesticated rice. From this large dataset, we retained 57 rice genotypes that had no missing data in the trait measurement. We then used this set of 57 rice genotypes, and randomly selected 20 genotypes to define drought-responsive

DEGs and used them as gene features for predicting the fecundity in the 37 "left-out" rice genotypes. We repeated this process ten times and the mean Pearson's $r$ was 0.62. The model performance was consistent across the evolutionarily distant Japonica and Indica rice sub-species (Supplementary Fig. 9 and Supplementary Data 9), and better than using the same number of random expressed genes (Mann–Whitney $U$-test, $P$-value $<2.2e-16$).

The mouse dataset comes from a highly genetically diverse Collaborative Cross (CC) population that comprises 90% of the genetic diversity across the entire laboratory Mus musculus genome[24]. The dataset we selected comprises matched transcriptome and disease outcome after influenza virus infection of 11 genotypes from the CC mouse population study[24]. We used DEGs (mock vs. infected) identified across the 11 mouse CC population genotypes to predict the disease outcome (asymptomatic vs. symptomatic) and found the mean Pearson's $r$ to be 0.98. The models built using cross-genotype DEGs outperformed the model using the same number of random expressed genes (Mann–Whitney $U$-test, $P$-value $= 3.3E-3$).

Overall, our results for the matched transcriptome and phenotype datasets for the rice and mice models provide two use-case studies of our evolutionarily informed machine learning pipeline applied to external data sets for traits in both plants and animals. They also show that transcript-based prediction can be achieved using a smaller population (20 and 11 genotypes in the case of rice and mice, respectively) compared with the requirement of hundreds of lines, which are needed for GWAS and eQTL studies[25].

**Validating the function of genes whose expression is influential in models predicting NUE.** The above studies established the robustness of our evolutionarily informed machine learning models in predicting trait outcomes based on conserved gene responses within and across species. Next, we experimentally validated gene features that are most influential in our predictive models. To this end, we used the feature importance score, an XGBoost[15] output, which reveals the influence of each feature (gene) in the predicted value (NUE) (Supplementary Data 3). We

reasoned that if models built for multiple genotypes selected a common set of gene features, this would indicate that those gene features are robust to genotype in predicting NUE. In maize, over 81% (202/248) of the XGBoost "important gene features" for predicting NUE were shared by models built for 16 genotypes, and 91% (245/248) were shared by ten or more maize genotypes (Supplementary Data 3). Similarly, for Arabidopsis 42% (257/610) of the "important features" for predicting NUE were shared by models built for 18 Arabidopsis accessions, and 85% (519/610) were shared by ten or more Arabidopsis accessions (Supplementary Data 3). These results are only consistent with the polygenic nature of NUE trait, but also reveal that there is a core set of influential N-DEGs, whose expression levels can accurately predict NUE phenotypes for both species.

In maize, the top-ranked "important gene features" in predicting NUE outcomes include the transcription factors (NLP, MYB, and WRKY), members of N-uptake/assimilation pathway (ammonium transporter, asparagine synthetase), and genes involved in photosynthesis and amino acid metabolism (Fig. 5, Step 4, Supplementary Data 3). In Arabidopsis, the top-ranked "important gene features" in predicting NUE include transcription factors (NF-Y, NLP, and MYB), members of the N-uptake/assimilation pathway (nitrate transporter, asparagine synthetase, and glutamine synthetase), tubulins, and chlorophyll *a-b* binding proteins (Fig. 5, Step 4, Supplementary Data 3). Several of the important features including the transcription factors (NLPs, LBD37/LBD38) and genes involved in N-metabolism (glutamine and asparagine synthetase) have been implied or directly linked to affect NUE in planta[21,26–29]. This consistency of our machine learning predictions of genes of "importance" to NUE with published results in planta not only validates the findings from our machine learning pipeline, but also indicates the novel genes uncovered in this pipeline could shed light on additional as yet unknown molecular components and mechanisms underlying NUE.

Further, we reasoned the regulatory genes (e.g., transcription factors, TFs) controlling the levels of expression of multiple XGBoost important features for predicting NUE would be ideal candidates for functional validation for their role in NUE in planta. To this end, we identified TFs predicted to regulate these XGBoost gene features of importance to NUE by constructing gene regulatory networks (GRNs) using GENIE3, which adopts the random forest machine learning algorithm and was the best performer in the DREAM4 and DREAM5 Network Inference Challenge[16].

To construct GRNs controlling NUE for each species, we first identified the N-responsive TFs in maize (545 TFs) and Arabidopsis (184 TFs) by intersecting the N-DEGs from our study with the TFs for each species using published databases[30–32]. Next, we used our N-response TFs in GENIE3 as the "regulatory genes" (GENIE3 term) whose influence on the evolutionarily conserved "target genes" in maize (248 gene features) or Arabidopsis (610 gene features) were weighed on a 0 to 1 scale, where 0 = non-influential and 1 = strongly influential. We kept the top 1% of the TF-target edges to construct the NUE regulatory network (Supplementary Data 4) and calculated the number of TF-target edges (connectivity) for each TF as a measure to evaluate their influence within the GRN.

Next, we integrated our GRN analysis with the XGBoost results to select candidate TFs that regulate genes of importance to NUE phenotype for functional validation of their role in NUE (Table 2). The selection and prioritization of TFs was based on one or more of the following criteria: (i) XGBoost-based importance score, (ii) GENIE3-based TF connectivity in the NUE GRN, (iii) curated knowledge from the literature, and iv) the availability of multiple mutant alleles. In Arabidopsis, the top TFs in the XGBoost-based

importance ranking listed in Table 2 include *NF-YA6* (AT3G14020), *DIV1* (AT5G58900), *UNE12* (AT4G02590), *NLP5* (AT1G76350), and *TCP2* (AT4G18390). The other two Arabidopsis TFs prioritized for *in planta* validation studies *WRKY38* (AT5G22570) and *WRKY50* (AT5G26170) (Table 2), were selected based on their high connectivity in the GENIE3-based GRN (Supplementary Data 4). For maize, we selected two candidate TFs (Zm00001d006293 *nlp17*, Zm00001d012544 *myb74*) for in planta validation studies that are hubs in the GENIE3-based GRN (Supplementary Data 4). Since no maize mutants were available for these genes, we took advantage of our cross-species approach by validating the function of their Arabidopsis homologs (AT1G76350 *NLP5*, AT5G06100 *MYB33*) in NUE. With the goal of cross-species validation, we also selected the maize homolog (Zm00001d006835, *NFYA3*) of the top-ranked Arabidopsis NF-YA6 (AT3G14020) for validation in NUE (Table 2). This choice took into consideration the fact that NF-Y transcription factors are enriched in Arabidopsis XGBoost gene features and in the maize GRN (Supplementary Data 3 and 4). Moreover, this selection was supported by previous studies which showed that overexpressing a member of the NF-YA family in wheat significantly increased N uptake and grain yield under different levels of N supply[33]. To discern the function of maize NF-Y homologs in NUE, we characterized the *NFYA3-1::UfMu* mutation with a Uniform Mu transposon insertion (mu1003041)[34] that does not produce a detectable full-length transcript.

Our results on the eight Arabidopsis TFs selected for in planta validation studies were classified into two groups based on our NUE phenotypic results (Fig. 6). The Group I "important gene features" in predicting NUE in Arabidopsis include *MYB33* (AT5G06100) and *TCP2* (AT4G18390), which when mutated showed increased NUE phenotypes under both high-N and low-N inputs (Fig. 6a). These validation results reveal that each TF plays a non-redundant role as negative regulators of NUE, as the loss-of-function T-DNA mutants displayed higher NUE under both N-deplete and N-replete conditions. The Group II "important gene features" in Arabidopsis include six TFs, which when mutated show increased NUE phenotypes specifically under low-N input: *UNE12* (AT4G02590), *NLP5* (AT1G76350), *NF-YA6* (AT3G14020), *WRKY38* (AT5G22570), *WRKY50* (AT5G26170), and *DIV1* (AT5G58900) (Fig. 6b). These validation results reveal that each of these Group II TFs plays a non-redundant role as negative regulators of NUE, as the loss-of-function T-DNA mutants displayed higher NUE, specifically under N-deplete conditions (Fig. 6b and Supplementary Fig. 10), suggesting that the function of these TFs in regulating NUE is only required when N is limiting. Alternatively, their function may be redundant with other TFs under N-replete conditions. For maize, the NUE tests of the *NFYA3-1::UfMu* mutant in the field showed that they accumulated less stalk and total N compared to wild-type, yet grain biomass, and all other traits dependent on grain biomass (grain yield, harvest index, NUtE) increased when grown with sufficient N (Fig. 6c). These results show that loss of maize NFYA3 influences how developing seeds sense and respond to plant N status, with the mutation reducing the N requirement to promote grain, thereby enhancing the NUtE. Observing phenotypes in the grain is also consistent with the expression pattern of *NFYA3*, which is strongest in developing seeds[35]. No significant differences were observed for NUE traits compared to wild-type maize (W22) when grown under N-limiting conditions, except for slightly lower grain yield and higher grain N concentration (Supplementary Data 5).

Taken together, our evolutionarily informed machine learning predictions of genes of importance to NUE and validation results for TF mutants for both Arabidopsis and maize demonstrate that:

**Table 2 Candidate TFs identified from XGBoost feature importance ranking for predicting NUE and/or hubs in GENIE3 network constructed from XGBoost important gene features.**

| Gene ID | Symbol | Published functions | Selection criteria |
|---|---|---|---|
| AT3G14020 | NF-YA6 | Male gametogenesis, embryogenesis, seed morphology, and seed germination; ABA response[42], NF-YAs are predicted target of miR169[45] | At XGBoost gene-to-trait model |
| AT4G02590 | UNE12 | Temperature-responsive SA immunity regulator[47] | At and Zm XGBoost gene-to-trait model |
| AT5G58900 | DIV1 | Nitrogen-response gene in the Arabidopsis seedling root and shoot[40] | At and Zm XGBoost gene-to-trait model |
| AT4G18390 | TCP2 | MicroRNA-mediated leaf morphogenesis[46], photomorphogenesis in Arabidopsis[50] | At XGBoost gene-to-trait model |
| AT5G22570 | WRKY38 | Basal defense[48] | At GENIE3 GRN |
| AT5G26170 | WRKY50 | Systemic Acquired Resistance[49] | At GENIE3 GRN |
| AT5G06100 | MYB33 | The Arabidopsis (MYB33), maize (Zm00001d012544) and rice (OsGAMYB) homologs are predicted target of miR159[44], juvenile-to-adult transition[44], anther development[43] | Zm GENIE3 GRN, At and Zm XGBoost gene-to-trait model, conserved cross-species function in anther development |
| AT1G76350 | NLP5 | The maize homolog of NLP5 (Zm00001d006293) is a marker for N status[19] and nutrient uptake[41] | Zm GENIE3 GRN, At and Zm XGBoost gene-to-trait model |
| Zm00001d006835 | NFYA3 | Photoperiod-dependent flowering and abiotic stress responses[51] | At XGBoost gene-to-trait model |

Our validation results confirming the roles of these eight TFs in NUE are provided in Fig. 6, Supplementary Fig 9, and Supplementary Data 5.

(i) Using evolutionarily conserved gene response significantly enhances the ability of the XGBoost machine learning models to predict NUE outcome across genotypes and species (plants and animals), and (ii) The XGBoost-based important scores and GENIE3-based connectivity are informative in selecting functionally important features—including TFs—to control of a complex physiological trait in crops—NUE—which has important implications for sustainable agriculture.

## Discussion

Our work addresses the primary, but often elusive, goal of genome-to-phenome analysis—namely, predicting phenotypic outcomes from genome-wide expression data. We show that exploiting evolutionary conserved gene expression datasets—within and across species—enhanced the machine learning model performance in predicting NUE phenotypes in a model (Arabidopsis) and a crop (maize), and also as applied to published matched transcriptome/phenotype datasets from another crop (rice) and model animal (mouse).

Our evolutionarily informed three-step machine learning pipeline (Fig. 1) which integrates phenotypic traits, transcriptome profiles, genetic variation, and environmental responses allowed us to; (1) preselect a subset of transcripts based on an evolutionarily conserved transcriptome responses within and across species, (2) employ this conservation as a biologically-principled way to reduce the feature dimensionality to improve the machine learning mmodel performance, and (3) rapidly validate the function of 'important gene features' identified from XGBoost models and GENIE3 gene regulatory network via the inclusion of a model and crop species.

The implementation of machine learning in predicting phenotypes has advanced in the past few years. However, the available datasets do not always; (1) exploit the genetic diversity of the organism(s) and (2) measure the phenotypes using same samples from which the transcriptome response was captured. Our work advances the field in both points, as we utilized a panel of genotypes with diverse genetic backgrounds and measured phenotypes from the same batch of plants that the transcriptome was captured. We integrated genetic diversity, machine learning, and cross-species approaches to identify genes of importance to an agronomically important trait, NUE. The trait we selected for study on NUE has the challenge of its underlying polygenic nature and the difficulty in collecting high quality phenotypic

data[36]. To this end, we designed a sufficiently large, but manageable experimental space of N-treatments across a set to ~20 genotypes spanning NUE phenotypes in a model and crop species. Our results presented herein, generated the largest matched phenotypic and transcriptomic datasets from both a model and a crop species. This dataset includes a large NUE phenotypic dataset resource of 318 maize genotypes for the plant community, and for 18 Arabidopsis accessions. We exploited the genetic diversity in 18 Arabidopsis accessions and 23 maize genotypes selected for broad phenotypic variation in NUE, and scored them for both transcriptomic and physiological responses in the same samples. Importantly, the selected maize genotypes represent the range of NUE diversity observed among a comprehensive collection of germplasm adapted to the U.S. Corn Belt, as confirmed empirically in the first year of our experiment (Supplementary Fig. 2 and Supplementary Data 1).

To extend this analysis beyond our proof-of-principle study of NUE, we applied our evolutionarily informed machine learning approach to other agricultural traits (e.g., drought resistance) in another major crop, using published transcriptome and phenotype datasets of genetically diverse rice subspecies (Indica and Japonica)[23]. In our application to animals, we exploited the growing awareness that host genetic variation has a major impact on pathogen susceptibility. To this end, we used matched transcriptome and phenotype data from a highly genetically diverse Collaborative Cross (CC) population that comprises 90% of the genetic diversity across the entire laboratory Mus musculus genome[24]. Models that we built using cross-genotype DEGs from both these studies of these genetically diverse lines in plants (rice) and animals (mice) lines, significantly outperformed the model using the same number of random expressed genes. Importantly, in these two additional case studies, and in our proof-of-principle example, our evolutionary informed analysis of matched transcriptome and phenome data allowed us to use a considerably smaller sample size compared to those needed for GWAS or eQTL studies[25].

The limitations of our study lie in the fact that the predictive models we derived do not necessarily inform the gene-to-trait causal relations. However, predictive accuracy and explanatory power are two dimensions rather than extremes in deciphering the complexity of the underlying mechanism[37]. Predictive modeling forecasts new or future observations, while explanatory modeling tests causal explanations[37]. Predictive models do not

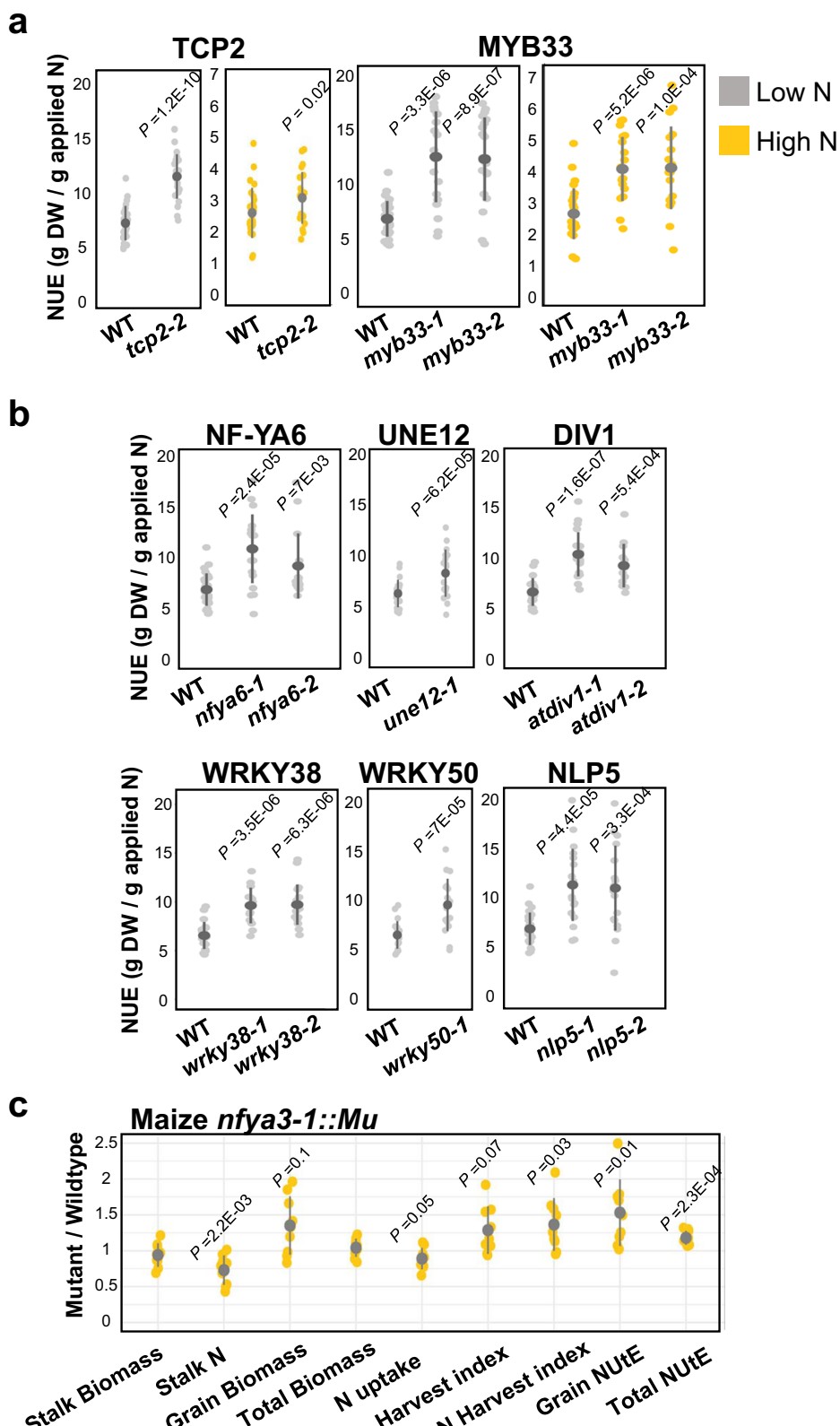

aim to explain the underlying mechanism; however, "the soundest path is to go for predictive accuracy first, then try to understand why"[38]. By providing accurate prediction, the predictive models reveal novel gene features for further investigation of causality[37]. We prove this principle using a reverse genetics approach to validate the function of eight transcription factors important to predicting NUE outcomes (Table 2). Notably, our

two-way cross-species validation strategy enabled us to verify the function of genes involved in NUE for (i) two maize candidate genes using mutants in their Arabidopsis homologs and (ii) one Arabidopsis candidate TF via analysis of a mutant in its maize homolog grown in the field (Table 2 and Fig. 6). The former is particularly important, especially when knockout mutants are less comprehensive for crop species. The use of a model species

**Fig. 6 Experimental validation of candidate TFs in NUE using loss-of-function mutants for Arabidopsis (lab) and maize (field). a** The Arabidopsis T-DNA mutants (Methods) in Group I genes displayed higher NUE compared to wild-type under N-replete (yellow, 10 mM $KNO_3$) and N-deplete (gray, 2 mM $KNO_3$) conditions. This suggests their non-redundant role(s) in regulating NUE regardless of the environmental N levels. **b** The Arabidopsis mutants in Group II genes displayed higher NUE specifically under N-deplete conditions. This indicates that the Group II genes are either only required under N-deplete conditions or are functionally redundant under N-replete conditions. The experiments were caried out three times with ten or more plants per genotype per condition. **c** Changes in NUE and component traits for the maize *nfya3-1::Mu* mutant compared to wild-type W22. Plants were grown in the field supplied additional N (150 kg N fertilizer/ha). Trait values are the average of five plants sampled from each of three replicate field plots, 15 plants per genotype (Methods). The higher total NUtE observed in the mutant was a combinatorial effect of lower stalk N (g/plant) ($P = 0.002$), total N uptake ($P = 0.05$) and higher grain biomass ($P = 0.1$). The increased NUE phenotype was also observed in the Arabidopsis T-DNA mutant defective the homolog gene NF-YA6 (AT3G14020) (**b**). The pointrange indicates mean $+/-$ SD. The *P*-value was calculated between WT and indicated mutant allele using one-sided *t*-test with unequal variance. The source data is provided in Supplementary Data 5.

provides genetics tools for functional characterization of candidate genes, given the fact that transforming Arabidopsis is far more efficient than in any crop species.

One interesting result we obtained, was that the learned model performance is more robust to maize genotype, compared with the models learned in Arabidopsis (Fig. 5). This outcome was obtained even though the maize genotypes selected for our study possess greater genetic diversity of NUE (Fig. 3c). Many factors may contribute to this difference. For instance, the maize gene features were applied to forecast NUE traits measured at later development stages (Supplementary Fig. 1). By contrast, the Arabidopsis gene features were applied to predict the NUE traits measured at the same time as RNA samples (Supplementary Fig. 1). An alternative, and not mutually exclusive explanation, comes from the fact that the maize leaf samples were collected from a specific developmental window with peak N-responsiveness[39], resulting in the apparent higher N-sensitivity, compared to the Arabidopsis samples. Our results support the hypothesis that pinpointing a developmental stage when the transcriptome response is most sensitive to the factor of interest (e.g., N-treatment in our study), would be beneficial to the performance of predictive models for the trait.

Our findings highlight the fact that genes affecting NUE are involved in an array of processes (Table 2), including nutrient response and uptake (*DIV1*[40] and *NLP5*[21,41]), anther and pollen development (*NF-YA6*[42] and *MYB33*[43]), juvenile-to-adult transition (*MYB33*[44]), microRNA-mediated growth and responses (*NF-YA*[45], *MYB33*[44], and *TCP2*[46]), immune response (*NF-YA6*[42], *UNE12*[47], *WRKY38*[48], and *WRKY50*[49]), and photomorphogenesis (*TCP2*[50] and Zm00001d006835[51]). These results not only provide additional evidence supporting the notion that NUE is a polygenic trait and intertwined with diverse signaling pathways, but further reveal a novel role of these genes in regulating NUE. Notably, there are three transcription factor families, NF-Y, NLP, and WRKY, whose members are enriched as the gene features of XGBoost models and/or the regulators of GENIE3-based GRN:

Our results identified nine Arabidopsis and one maize *NF-Y* genes as the features in XGBoost models, as well as 12 Arabidopsis and 14 maize *NF-Y* genes, as potential regulators in the GENIE3 NUE GRN (Supplementary Data 3 and 4). Moreover, we validated the function of *NF-YA6* in NUE—a top gene in Arabidopsis XGBoost model—using mutants in Arabidopsis *NF-YA6* (AT3G14020), as well as its maize homolog *NFYA3* (Fig. 6). The NF-Y family, found in nearly all eukaryotes[52], encodes components of an evolutionarily conserved trimeric transcription factor complex. In humans, NF-Y binds to the CCAAT box in promoters of large sets of genes overexpressed in breast, colon, thyroid, and prostate cancer[53]. In plants, the regulatory roles of NF-Y have been revealed in flowering-time, early seed development, nodulation, hormone signaling, and stress responses[52]. NF-Ys function as a multimeric protein complex (NF-YA/B/C(-CO/bZIP/bHLH) to bind its canonical motif CCAAT and/or the

motif(s) of its partner TFs[54]. It is tempting to hypothesize that the flexible cis-binding capacity makes NF-Ys versatile and context-dependent TFs that can quickly adapt to nutrient fluctuations. It is noteworthy that several *NF-Y* genes are targeted and down-regulated by miR169[55] and miR169 members respond transcriptionally to N-starvation[56]. Thus, our data supports a new link between N-signaling, miRNA changes in N-responsive of NF-Ys, to the phenotypic output of NUE: Nitrogen → miR169 → NF-Y → NUE.

We identified six Arabidopsis and two maize *NLP* genes as the features in XGBoost models to predict NUE, as well as five Arabidopsis and 14 *NLP* genes as potential regulators in the GENIE3 NUE GRN (Supplementary Data 3 and 4). Further, using mutants, we validated the role of *NLP5*—a top gene feature in maize XGBoost model and maize NUE GRN—as a negative regulator of NUE specifically under low-N conditions (Fig. 6b and Supplementary Fig. 9). The NLPs—which are plant-specific TFs—are related to a core symbiotic gene *Nin*[57] and later identified as master regulators of nitrate signaling in Arabidopsis[26]. Emerging evidence suggests their contribution to N-regulated gene expression and developmental processes is common across plant species[58]. The results from our functional validation experiment indicated that *NLP5* is a negative regulator of NUE under N-depleted conditions (Fig. 6b), which can be explained by the fact that *NLP5* is a target of NIGT1/HRS1, a master regulator of N-starvation response genes[59,60]. Thus, the loss of *NLP5* in the Arabidopsis mutants could de-repress the N-starvation response, leading to higher NUE.

We identified six Arabidopsis and six maize *WRKY* genes as the features in XGBoost models, as well as 24 Arabidopsis and 11 *WRKY* genes as the regulators in GENIE3 NUE GRN (Supplementary Data 3 and 4). Among them, WRKY38 and WRKY50 are the top-ranked TF hubs in the Arabidopsis NUE GRN. Our functional analysis using Arabidopsis mutants validated a role of WRKY38 and WRKY50 in mediating NUE (Fig. 6b). WRKYs, occurring primarily in plants[61], are among the largest families of transcription factors. Cumulative evidence has demonstrated the important biological functions of WRKYs in plant developmental processes (embryogenesis, germination, senescence etc.) as well as response to biotic and abiotic stresses including defense, salt, drought, nutrient starvation, and more[62]. In addition to their known functions in defense responses[48,49], our results add a novel aspect to *WRKY38* and *WRK50* in regulating NUE and make them candidate TF hubs in coordinating plant responses to N levels as well as biotic stress.

Our work demonstrates that the integration of genetic diversity, cross-species transcriptome analysis, and machine learning method enhances predictive modeling of genes affecting NUE. Our results from reverse genetic analysis further show that those genes predictive of NUE are not only biomarkers, but also are functionally important in determining plant performance in response to environmental nutrition. The pipeline presented in

this work could complement the current approaches in identifying important genes in a multigenic trait. Our validation of the evolutionarily informed strategy for feature reduction across both genetically diverse crop and animal datasets, supports its potential to inform any system that seeks to uncover important genes controlling a complex phenotype in biology, agriculture, or medicine.

## Methods

### Plant materials, growth conditions, and phenotypic assays

*Arabidopsis*. All Arabidopsis seeds used in this study were obtained from ABRC (https://abrc.osu.edu/) and listed in Supplementary Data 1. The 18 Arabidopsis accessions are Akita, Bl-1, Bur-0, Col-0, Ct-1, Edi-0, Ge-0, Kn-0, Mh-1, Mr-0, Mt-0, N13, Oy-0, Sakata, Shahdara, St-0, Stw-0, and Tsu-0, as previously studied for NUE[8]. The T-DNA mutants are all in the Col-0 background. The mutant lines[63] are *myb33-1* (SALK_056201), *myb33-2* (SALK_065473), *tcp2-2* (SALK_060818), *une12-1* (SAILseq_711_E09.1), *nlp5-1* (SALK_055211), *nlp5-2* (SALK_063304), *nfya6-1* (SALK_005942), *nfya6-2* (SAIL_159_E03), *wrky38-1* (WiscDsLox489-492C21), *wrky38-3* (SAIL_749_B02), *wrky50-1* (SAIL_115_C10), *div1-1* (SALK_056735), and *div1-2* (SALK_084867C). The mutants were genotyped to confirm the homozygosity. The expression level of the inserted gene in the homozygous mutants were below detection limit of real-time PCR (Supplementary Fig. 11). The primer sequences are provided in the Supplementary Data 6.

For growth experiments, the Arabidopsis seeds were germinated on ½ MS with MES Buffer and Vitamins (RPI cat M70800) plates for 7–10 days in on a 16h-light/8h-dark photoperiod. The seedlings were then transferred to pre-washed nutrient-poor matrix vermiculite under an 8 h light (120/μmol²/s)/16 h dark diurnal cycle, at temperatures 22 and 20 °C respectively and 40% humidity. We kept one plant per pot and carried out the entire experiment using Arasystem (https://www.arasystem.com/). To track the N supply for each plant, we treated each plant with the same amount of low N (LN, 2 mM KNO₃) (Sigma cat P6083) or high N (HN, 10 mM KNO₃) medium (Caisson Labs cat. no. MSP10) using a syringe and recorded the volume. The potassium concentration was maintained by supplementing KCl (Sigma cat P9333) to the LN medium. On 40 and 42 DAS, the treatment was enriched with 10% atom excess ¹⁵N for ¹⁵N influx analysis. To minimize the variation due to pot location in the growth chambers, the HN row was located adjacent to the LN row, and the flats were shuffled three times weekly. We repeated these experiments three times consecutively to obtain biological replicates for phenotypic and transcriptomic samples. For each of the 18 Arabidopsis accessions, mature leaves were harvested for transcriptome and the above ground tissues for physiological traits at 43 DAS. The dried tissues were ground and analyzed for total nitrogen using a PDZ Europa ANCA-GSL elemental analyzer interfaced to a PDZ Europa 20–20 isotope ratio mass spectrometer at UC Davis Stable Isotope Facility. The phenotypic data are provided in Supplementary Data 1.

*Maize*. Seeds for all maize inbreds used in this study were originally obtained from the USDA-ARS North Central Plant Introduction Station in Ames, IA, except for the inbreds derived from the Illinois Selection Experiment and FR1064 as described in Uribelarrea et al.[23]. Inbred lines were subsequently increased by controlled self-pollination, and hybrid seed produced by controlled crosses. We grew the maize plants in N-managed field plots in Urbana, Illinois between May and September in 2014–2016. The soil type is a Drummer silty clay loam, pH 6.2, that received either 200 kg/Ha fertilizer N or no exogenous applied N when the plants reached the V3 growth stage. Subsequent soil testing and measures of plant N recovery estimate approximately 60 kg N/ha were made available from the soil alone. The N fertilizer was applied as granular ammonium sulfate banded adjacent to plants at the soil surface. Plants were grown in a split-plot design, where individuals in each main plot (2 rows 5.3 m long, 76 cm row spacing) were paired in adjacent rows of N-replete or N-depleted condition to a final density of 49,000 plants per hectare for inbreds and 77,000 plants per hectare for hybrids. Genotypes within main plots were arranged by relative maturity to minimize its impact on NUE traits. Plots were maintained weed free by a pre-plant application of herbicide (atrazine + metalochlor) followed by hand weeding as needed.

Maize phenotyping was performed at the R6 growth stage, when plants have reached physiological maturity, but may not yet have fully senesced. Five plants from each plot were cut at ground level, ears removed, and a fresh weight obtained on the entire remaining plant material (stover, comprising mostly stalk by weight, followed by leaves, tassels, and husks). The stover was then shredded in a Vermeer wood chipper, a subsample was collected into a tared cloth bag, and the subsample fresh weight was recorded. Stover samples were oven-dried to dryness at least three days at 65 °C and the subsample dry weight used to estimate stover biomass. The dried stover was further ground in a Wiley mill to pass through a 2 mm screen, and approximately 100 mg used to estimate total nitrogen concentration by combustion analysis with a Fisons EA-1108 N elemental analyzer. Grain samples were dried for approximately one week at 37 °C, after which grain was shelled from the cobs, and the cob weight recorded. The moisture content and N concentration within each 5-plant grain sample was estimated using near-infrared reflectance spectroscopy on a Perten DA7200 analyzer, using a custom calibration built with samples possessing a broad range of variation in composition and color. The

nitrogen concentration calibration was established using data from total combustion analysis of grain samples as described above for stover.

The *NFYA3-1::Mu* loss-of-function allele was generated by the UniformMu insertion mu1003041::Mu in the 5′ untranslated region the annotated gene model Zm00001d006835. The UFMu-00332 seed stock was obtained from the Maize Genetics Cooperation Stock Center and genotyped[64] to identify homozygous for the *NFYA3-1::Mu* mutant allele, which were then self-pollinated. The expression level of the *NFYA3* gene in the homozygous mutants was below detection limit of real-time PCR (CT > 45) (Supplementary Fig. 10). The primer sequences are provided in the Supplementary Data 6. The *NFYA3* mutant and wildtype W22-UniformMu plants were grown in 2020 at the same field site and using the same experimental design, nitrogen treatments, and phenotyping methods described above. The phenotypic data are provided in Supplementary Data 5.

### RNA extraction, library preparation, and sequencing

For each of three Arabidopsis RNA replicates, we harvested mature leaves from pre-bolting plants on 43 DAS between 9 and 11 AM from two plants, flash froze in liquid nitrogen and stored in −80 °C. We isolated RNA using Direct-zol RNA Kits following manufacturer's instructions (Zymo Research). RNA quality was assessed on an Agilent Tape station using RNA ScreenTape (Agilent cat 5067-5576). All 108 stranded RNA-seq libraries were made using the NEBNext® Ultra™ II Directional RNA Library Prep Kit for Illumina® (NEB cat E7768) and assessed using DNA high sensitivity D1000 ScreenTape system (Agilent cat 5067-5584). The RNA-Seq libraries were sequenced using Illumina HiSeq 2500 v4 with 1 × 75 bp single-end read chemistry at the GenCore Facility at New York University Center for Genomics and Systems Biology.

For each of three maize RNA replicates, we collected leaf tissues from two inches from the base of leaf 13 subtending the top ear at R1 stage between 9 and 11 AM, flash froze in liquid nitrogen and stored in −80 °C. We extracted RNA from frozen leaf tissue using CTAB-chloroform method. Genomic DNA was removed using DNAse I (NEB cat M0303). RNA-seq libraries were prepared using a TruSeq Stranded mRNAseq Sample Prep kit (Illumina cat RS-122-2101) according to the protocol provided. Single-end 150 bp reads were generated using the Illumina HiSeq 4000 at the Roy J Carver Biotechnology Center in the University of Illinois at Urbana-Champaign.

### Identification of N response differentially expressed genes (N-DEGs)

All RNA-seq raw reads were processed using the same pipeline to remove optical duplicates (Clumpify 37.24) and adapters (BBDuk 37.24)[65]. The trimmed reads were aligned to the latest genome in 2018, TAIR10[66] for Arabidopsis and Zm-B73-REFERENCE-GRAMENE-4.0[14] for maize, using BBMap (37.24). The mapped reads were assigned by featureCounts (1.5.1)[67] using the latest annotation in 2018: Araport11[68] for Arabidopsis and AGPv4.32[14] for maize. The parameters and software versions for the above steps are available in Supplementary Data 7 and GEO accession GSE152249. We identified N-DEGs in the training data set ($n{-}1$ genotypes) and repeated $n$ times ($n$ = number of genotypes in each species). In each round of analysis, we first filtered out the lowly expressed genes (CPM > 1 in less than ten samples) and normalized the data using upper-quantile (EDASeq 2.18.0)[69] and replicate samples (RUVSeq 1.18.0)[70]. Subsequently, we used edgeR (3.26.8)[19] to detect genes differentially expressed in high vs low N condition across genotypes (FDR < 0.05). Lastly, we intersected the $n$ lists of DEGs and only retained the ones occurring on $n$ lists as a common set of N-DEGs. These analyses resulted in 2,123 Arabidopsis N-DEGs and 6914 maize N-DEGs (Fig. 4). The Arabidopsis-Maize homolog mapping file is generated from Phytozome 10[18] and available in Supplementary Data 8. The code is available in the Coruzzi lab Open Science Framework (https://osf.io/avjph/).

We held out a testing genotype before the DEG stage; and only training genotypes (n-1 genotypes) were used in DEG analysis and XGBoost models (as depicted in new Fig. S4). The held-out test genotypes were then used to validate the model performance. This round robin approach (Fig. S4a(i) and b(i)), generated 18 and 16 independent DEG lists for Arabidopsis and maize, respectively. In approach a, we identified a unified list of gene features by intersecting these independent lists (e.g., 18 for Arabidopsis and 16 for maize) (Fig. S4a(ii)). By contrast, in approach b, cross species analysis was performed on each independent DEG list (e.g., 18 for Arabidopsis or 16 for maize).

To rule out the possibility that using the intersected DEGs (e.g., within species) would overly optimize the XGBoost results, we further compared the XGBoost performance using the intersected DEGs (Fig. S4a) with the alternative approach that did not go through the within species list intersection (Fig. S4b). The results of these two approaches are comparable (Fig. S4c). However, the advantage of conducting the cross-genotype intersection (Fig. S4a, which we used in this manuscript), has the benefit of resulting in a unified list of gene features, compared to multiple independent lists of gene features. Generating a unified list of gene features will enable the gene feature ranking across genotypes, rather than restricted to an individual genotype.

### Construction and evaluation of predictive machine learning models

We used a tree model with gradient boosting, XGBoost[15] R implementation, to train and test the models. For each species, we split the data into training ($n{-}1$ phenotypes) and testing (left-out genotype) sets. We used five-fold internal cross-validation to select

the optimized hyperparameters. We tuned "nrounds" (number of trees), "col-sample_bytree" (the proportion of features for constructing each tree), "sub-samples" (the portion of training data samples for training each additional tree), and "eta" (shrinkage of feature weights to make the boosting process more conservative and prevent overfitting) in an XGBoost:regression model. Subsequently, we made predictions on each of the left-out genotype, assessed the model accuracy by calculating the Pearson's correlation coefficient $r$ between the predicted and actual values[71], and reported the $r$ from 100 iterations.

**Selection of candidate genes for functional validation in NUE**. We used two parallel procedures to select candidate genes for functional validation. First, we used the XGBoost-generated feature importance score that indicates how useful each feature was in the construction of model. We summed the score on a gene-by-gene basis from 18 models for Arabidopsis and 16 models for maize and generated a ranked list (Supplementary Data 3). Second, we used a Random Forest-based algorithm GENIE3 to infer the transcription factors regulating the gene features. We used the N-responsive TFs (184 Arabidopsis TFs and 545 maize TFs) as the regulators and the gene features (610 Arabidopsis genes and 248 maize genes) as the targets and kept the default parameters. We constructed the NUE regulatory network using the top 1% of the edges and ranked the TFs based on their connectivity (number of edges) (Supplementary Data 4).

**Reporting summary**. Further information on research design is available in the Nature Research Reporting Summary linked to this article.

## Data availability

The raw and processed data generated in this study including FASTQ, BAM, and read counts have been deposited in the Gene Expression Omnibus (GEO) under accession code GSE152249. Source data are provided with this paper and specified in the legend of each figure. Source data are provided with this paper.

## Code availability

The code used in this study is available at the Coruzzi lab Open Science Framework (https://doi.org/10.17605/OSF.IO/AVJPH) [72].

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

## Acknowledgements

We would like to thank Viviana Araus, Matthew D Brooks, Manpreet S. Katari, Jacapo Cirrone, and Dennis Shasha for constructive discussions. We thank Katerina Holan for assistance in RNA extraction. We acknowledge members of the Coruzzi lab at NYU Biology and Shasha lab members from NYU Courant Institute for feedback and comment throughout the project. We acknowledge and thank the Arabidopsis Biological Resource Center (ABRC) and Maize Genetics Cooperation Stock Center for supplying the insertion lines. This work was supported by the National Science Foundation Plant Genome Research Program (IOS-1339362) to G.M.C., Y.L., and S.M., the USDA National Institute of Food and Agriculture Hatch project numbers 1013620 to Y.L., the USDA-NIFA predoctoral fellowship (2016-67011025167) and NSF CompGen fellowship to J.A., the Jonathon Baldwin Turner graduate fellowship from the College of Agriculture, Consumer, and Environmental Sciences at the University of Illinois at Urbana-Champaign to J.B.

## Author contributions

C.Y.C., Y.L., K.V., S.P.M. and G.M.C. conceptualized and designed research. C.Y.C., H.J.S., G.K. Justin Halim, S.H.P., H.Y.C. and G.L. conducted Arabidopsis lab experiments. J.A. and J.B. conducted the maize field experiments. C.Y.C. analyzed the data with input from Ji Huang, Y.L., K.V., S.P.M. and G.M.C. C.Y.C. prepared the figures and tables. C.Y.C. and GMC wrote the manuscript with input from Y.L., Ji Huang, J.A., and S.P.M.

## Competing interests

The authors declare no competing interests.
