## [Peer Review File · Nature Communications]

Evolutionarily informed machine learning enhances the power of predictive gene-to-phenotype relationshipsReviewers' Comments:

Reviewer #1:

Remarks to the Author:

Cheng et al describe the use of a novel targeted machine learning tool that is central to an investigative pipeline relying on transcriptional gene expression and phenotypic analysis to predict gene associations linked to the genetic signatures linked to nitrogen use efficiency in plants. The paper claims, that through the identification of evolutionary conserved N-responsive plant genes (gene expression that increases or decreases in response to N availability), machine learning tools can be used to train and create predictive models linking gene activity with plants demonstrating NUE traits. The claim also extends to any transcriptional/phenotypical relationship in biology, providing a quick and less demanding approach to identify/link gene signatures to physical traits – effectively linking cultivated transcriptional genome analysis to end-point phenomics.

In this study, Cheng et al used a NUE diversity set of *Arabidopsis thaliana* (model plant) alongside a corresponding diversity collection of inbred and hybrid *Zea mays* plants (crop plants) which show variation in NUE traits. Controlled provision of external N (low versus high N fertilisation rates) under controlled or field-based conditions allowed plants to grow on a defined N budget from which NUE calculations (N uptake relative to growth) and NutE (N utilisation efficiency) could be estimated across the two (model and crop) diversity sets, respectively. Corresponding transcriptional data sets were collected from tissues to link measured NUE traits to defined tissue gene expression patterns. Differentially expressed genes were identified between and across both plant systems that identified key evolutionary conserved gene signatures which responded in tandem between the model and crop species. These genes were used in a predictive model program (XGBoost) to identify which genes were best at predicting NUE traits and then explored for their gene regulatory networks (GRN). A number of transcription factor (TF) families were identified as potential regulators of the identified genes. A select number were verified using multiple gene KO studies in the model plant and one confirmation using a maize mutant.

This is a novel study employing a mixture of existing and new tools to help interpret transcriptional networks and verify if they can be used to effectively predict a defined phenotype. NUE is a polygenic trait in plants, involving a range of actors to coordinate N uptake, assimilation, redistribution and remobilisation to support growth and yield. No one gene has yet been shown solely responsible for improving NUE in plants although key TF's (e.g. NLP's) when overexpressed appear to have some influence on the trait in other independent studies. The identified TF families (NF-Y, NIN-like (i.e. NLP's) and WRKYs) identified in this study are great candidates for critical NUE regulatory control points. Indeed, loss of activity studies with a select few of these TF's created responses in *Arabidopsis* which mirror an influence to N utilisation and growth. A second novel aspect of this work is the parallel use of model and crop plants, where the model system is effectively used as a surrogate to test and define transcriptional signals in the more recalcitrant crop plant. This positive relationship is substantial to the plant science discipline, where transgenics in most other crop plant species where NUE has an impact is difficult and time consuming. The manuscript would benefit from extra levels of verification on the efficacy of this approach relative to other systems (GWAS or eQTL studies). Have similar studies employing GWAS analysis identified similar gene families? I would expect the conserved nature of the identified NUE genes should have been revealed to some extent in other studies linked to NUE?

Defining NUE traits in crop plants has been difficult due to the polygenic nature of the trait and the lack of effective plant resources in the major crop plants. This study demonstrates that common transcriptional approaches when aligned with careful phenotypic analysis can provide the basis for novel paths of gene discovery to help explain a specific trait. It is very useful as an interpretative tool though I'm not sure how practical the approach will be in advancing NUE selections in the short-term until the gene to function relationships are defined and marked.

Clarification:

- 1) The range of NUE variabilities in the maize inbred population is small relative to the hybrids and that across the Arabidopsis population. Have other more diverse lines (e.g. Buckler collection) also been tested to enhance or challenge this model? Some discussion on the selection of the lines in both systems would be welcome. Do the identified TF's come up in other plant species such as rice and Medicago?
- 2) The fact that many of the identified genes influencing NUE have links to other known processes in plants raises the question on how specific these genes are to NUE? Is there an estimate of the false positive rate for this type of analysis?
- 3) Using the same gene transcription data sets, have other traits been tested for such as seed yield, seed N and plant biomass? Do similar gene networks present themselves – i.e. how unique are these genes to NUE?
- 4) As NUE is a cumulative polygenic trait, did the authors test alternative Maize RNA samples which were not N-responsive as a control? Can this approach work without a specific responsive sampling time/tissue to help predict unknown traits?
- 5) An explanation on how this technique can be transferred to other systems and would be useful to back-up the last sentence of the abstract and in the discussion.

Reviewer #2:

Remarks to the Author:

The research work in this manuscript studies nitrogen use efficiency (NUE) in crop Maize by building a gene-to-trait model based on transcriptome analysis and a comparative study with Arabidopsis data. Experimental phenotypic and genomic data about NUE were collected and analyzed by a pipeline built with a collection of bioinformatics, statistical and machine learning tools. While the study generates valuable new phenotypic and genomic data for NUE phenotype and led to interesting findings of gene-to-trait models, there is a flaw in the design of data analysis pipeline and some important analyses are missing. The introduction section does not provide a useful literature review and related work is completely missing.

Major points:

1. The machine learning analyses were intended for both predictive and descriptive modeling, which is a strength of this study. While the result of descriptive modeling for selecting the NUE driver genes, the evaluation of the predictive modeling is inaccurate and unacceptable (e.g. results in Table 1 and Step 3 in Figure 5). In particular, The differential gene expression analysis with general linearized model already utilized phenotypic data (NUE output). Evaluating how the selected genes are predictive of the NUE output in a machine learning model is an over-optimistic evaluation since the output of the test set in the leave-one-out-crossvalidation has already been used to identify the genes. This leak of test label information is particularly problematic for evaluation on datasets of small size since some features could be selected to capture a particular sample in the small dataset.

Thus, the predictive modeling should be re-evaluated in a separate cross-validation. In particular, the cross-validation should be applied to the analysis in Fig 3 and Fig 5 together, i.e. a test sample is held out before the differential gene expressions analysis and the analysis should be repeated for every held-out sample. Without this rigorous evaluation, the predictive modeling is not validated as claimed.

2. While the choices of the methods such as generalized linear model and XGboost (random forests) seem to be reasonable, there is insufficient comparison with baselines or other commonly used alternative methods. In particular, the comparisons in Table 1 are not useful with randomly chosen genes or N-DEGs. How does the top genes selected by evolutionarily conserved N-DEGs compared with the same number of original top DEGs without using the evolutionary analysis? Other than generalized linear model, DEG analysis should also be compared with commonly used t-test and

Wilcoxon ranksum test. In addition, how different is the ranking by XGboost compared with the original ranking by p-values in statistical analysis after filtering with evolutionary information? This comparison is necessary to justify that XGboost captures features beyond single variable analysis.

3. The introduction section about machine learning and "curse-of-dimensionality" is too broad and irrelevant to this study given the audience of Nature Communication are from a broader community of many other research fields. There are many other machine learning methods and pipelines were proposed for comparative studies across species. For example, people use canonical correlation analysis (CCA) to co-project two different datasets. The contribution of this study is limited to plant species unless the authors include a comprehensive survey of (and a comparison with) those comparative genomics methods such as those many methods for integrative analysis of human and mouse transcriptomes.

Minor points:

1. The analysis in Fig. 2 is still confusing. What are the phenotypic features used in SVD? What's their scale (mean and variance)? It appears Nitrogen use is not a feature used in SVD analysis of Arabidopsis data, which should be clarified. The variance of each ET should also be provided. ET1-ET3 should also be visualized to show the relevance to the phenotypic features.

2. In Figure 3, the superscript 1 and 2 above Feature Reduction is unannotated.

3. The heat map of DEG expressions in Figure 4 is not properly annotated and not helpful for the understanding of the research work. If there is not enough space to show the gene names, this figure should be moved to supplementary document, a figure showing a subset of most interesting genes can be included.

4. In addition, since there is no evidence or comparison with other similar data integration methods, the last statement (line 67-69) on page 2 is not well supported by this study.

5. The sentence between line 48-49 on page 2 seems not to be logical after the previous paragraph about "curse-of-dimensionality". Why exploiting transcriptome data to predict nitrogen use efficiency can address the afore mentioned computational problems?

Reviewer #1 (Expertise: Nitrogen use in plants)

Cheng et al describe the use of a novel targeted machine learning tool that is central to an investigative pipeline relying on transcriptional gene expression and phenotypic analysis to predict gene associations linked to the genetic signatures linked to nitrogen use efficiency in plants. The paper claims, that through the identification of evolutionary conserved N-responsive plant genes (gene expression that increases or decreases in response to N availability), machine learning tools can be used to train and create predictive models linking gene activity with plants demonstrating NUE traits. The claim also extends to any transcriptional/phenotypical relationship in biology, providing a quick and less demanding approach to identify/link gene signatures to physical traits – effectively linking cultivated transcriptional genome analysis to end-point phenomics.

In this study, Cheng et al used a NUE diversity set of *Arabidopsis thaliana* (model plant) alongside a corresponding diversity collection of inbred and hybrid *Zea mays* plants (crop plants) which show variation in NUE traits. Controlled provision of external N (low versus high N fertilisation rates) under controlled or field-based conditions allowed plants to grow on a defined N budget from which NUE calculations (N uptake relative to growth) and NutE (N utilisation efficiency) could be estimated across the two (model and crop) diversity sets, respectively. Corresponding transcriptional data sets were collected from tissues to link measured NUE traits to defined tissue gene expression patterns. Differentially expressed genes were identified between and across both plant systems that identified key evolutionary conserved gene signatures which responded in tandem between the model and crop species. These genes were used in a predictive model program (XGBoost) to identify which genes were best at predicting NUE traits and then explored for their gene regulatory networks (GRN). A number of transcription factor (TF) families were identified as potential regulators of the identified genes. A select number were verified using multiple gene KO studies in the model plant and one confirmation using a maize mutant.

This is a novel study employing a mixture of existing and new tools to help interpret transcriptional networks and verify if they can be used to effectively predict a defined phenotype. NUE is a polygenic trait in plants, involving a range of actors to coordinate N uptake, assimilation, redistribution and remobilisation to support growth and yield. No one gene has yet been shown solely responsible for improving NUE in plants although key TF's (e.g. NLP's) when overexpressed appear to have some influence on the trait in other independent studies. The identified TF families (NF-Y, NIN-like (i.e. NLP's) and WRKYs) identified in this study are great candidates for critical NUE regulatory control points. Indeed, loss of activity studies with a select few of these TF's created responses in *Arabidopsis* which mirror an influence to N utilisation and growth. A second novel aspect of this work is the parallel use of model and crop plants, where the model system is effectively used as a surrogate to test and define transcriptional signals in the more recalcitrant crop plant. This positive relationship is substantial to the plant science discipline, where transgenics in most other crop plant species where NUE has an impact is difficult and time consuming. The manuscript would benefit from extra levels of verification on the efficacy of this approach relative to other systems (GWAS or eQTL studies). Have similar studies employing GWAS analysis identified similar gene families? I would expect the conserved nature of the identified NUE genes should have been revealed to some extent in other

studies linked to NUE?

Defining NUE traits in crop plants has been difficult due to the polygenic nature of the trait and the lack of effective plant resources in the major crop plants. This study demonstrates that common transcriptional approaches when aligned with careful phenotypic analysis can provide the basis for novel paths of gene discovery to help explain a specific trait. It is very useful as an interpretative tool though I'm not sure how practical the approach will be in advancing NUE selections in the short-term until the gene to function relationships are defined and marked.

We thank Reviewer #1 for acknowledging the novelty and contribution of this manuscript. We have clarified/addressed each point raised by the reviewer in the revised manuscript, as detailed below.

Reviewer 1 requested Clarifications/Revisions:

1) The range of NUE variabilities in the maize inbred population is small relative to the hybrids and that across the Arabidopsis population. Have other more diverse lines (e.g. Buckler collection) also been tested to enhance or challenge this model? *Some discussion on the selection of the lines in both systems would be welcome. Do the identified TF's come up in other plant species such as rice and Medicago?*

Responses:

1.1 Some discussion on the selection of the lines in both systems would be welcome.

Response: We have clarified in the manuscript the rationale for selecting the genotypes from larger populations to identify a *smaller* set of genotypes representing a broad NUE diversity (**Lines 79-92**). For Arabidopsis, the 18 accessions we selected have been previously identified to have a broad range of NUE phenotypes, as described by Masclaux-Daubresse lab and other citations in the NUE literature (North et al., 2009; Chardon et al., 2010; Ikram et al., 2011; Masclaux-Daubresse and Chardon, 2011). These 18 selected Arabidopsis genotypes originate from a nested collection of 265 accessions found in a wide range of habitats differing notably in soil richness (McKhann et al., 2004). For maize, the genotypes were selected from a population of 318 maize genotypes including elite germplasm from different breeding programs with recently expired plant variety patents, Illinois Protein Strains, tropical and temperate lines (McMullen et al., 2009; Beckett et al., 2017). In the revised manuscript, we have now added the NUE data we collected for this population of 318 maize genotypes in **New Supplementary Table 1**, which we believe will be a valuable new resource for the maize community.

To further address this point, we have now quantified and reported the range of NUE variation in the genotypes in the revised manuscript. Specifically, we calculated the coefficient of variation (CV) to show the NUE variation for both Arabidopsis and maize (**New Figs 2a, 3a and S2**). Further, we included additional data to show that, when grown in the same plots with sufficient N in the field, the 12 maize inbreds selected for our study exhibited a similar coefficient of variation for NUE phenotypic values (CV = 0.19), as the larger population of 318 maize genotypes (CV = 0.15) (**New Supplementary Table 1**). These 12 maize inbreds, as well as their hybrids with B73, were used in our study for transcriptome profiling and detailed phenotyping in N-responsive field plots over three seasons. The CV of maize inbred in high-N and low-N field plots (CV = 0.50) is in fact higher than that of hybrid (CV = 0.23), both of which are lower than that of Arabidopsis (CV = 0.58). These new results are shown in the **New Figures 2a, 3a, and S2**, and described in the revised text, **Lines 115-117, 125-135**.

1.2 Do the identified TFs come up in other plant species such as rice and Medicago?

Response: The eight TFs identified and validated as having a role in NUE in our study, have previously been reported to function in various pathways and responses, as summarized in **Table 2**. As such, our results have uncovered TFs that link NUE to pathways previously associated with environmental and/or developmental responses. Several of the TFs we have identified as being involved in NUE in maize and Arabidopsis (e.g., NF-YA and NLP5, and DIV1), have also been previously linked to N-responses in other species. For example, we cite that overexpression of a wheat NF-YA significantly increased N uptake and grained yield under different N-supply levels (Qu *et al*, 2015) (**Lines 309-310**). *DIV1* (Arabidopsis N signaling) and NLP5 (Maize biomarker of N status, N uptake) are involved in general N response (**Lines 389-393**).

2) The fact that many of the identified genes influencing NUE have links to other known processes in plants raises the question on how specific these genes are to NUE? Is there an estimate of the false positive rate for this type of analysis?

Response: NUE involves uptake, assimilation, translocation, recycling, and remobilization. Each process is intertwined with environmental (drought, defense) and developmental cues (flowering time, senescence, autophagy etc) (reviewed by Masclaux-Daubresse *et al.*, 2010). Thus, as anticipated, our results show that the TFs identified and validated in NUE our study are also linked to other processes (**Table 2**). Although there is not yet a gold standard to estimate the false positive rate for this type of analysis, we have built models for additional traits to explore the specificity of gene importance for each trait. This point is elaborated in the response 3.1 below.

3) Using the same gene transcription data sets, have other traits been tested for such as seed yield, seed N and plant biomass? Do similar gene networks present themselves – i.e. how unique are these genes to NUE?

Response to these three questions are below:

3.1 Can these gene be used to predict additional traits?

3.1 To address this question, in the revised manuscript, we constructed models for additional traits, which is summarized in the table below and shown in the **New Figure S5** and discussed in the text (**Lines 228-243**). Our new analysis indicates that the genes in our analysis and our pipeline can be applied on more than one trait.

Species – Trait	Arabidopsis – Biomass	Arabidopsis – N uptake	Maize – Biomass	Maize – Grain Yield
Pearson's r	0.68	0.69	0.72	0.52

3.2 Are these gene features trait-specific? Do similar gene networks present themselves – i.e. how unique are these genes to NUE?

3.2 The XGBoost ranks gene feature importance to the trait, however, it does not construct gene regulatory networks. To address the specificity/uniqueness of the gene importance raised by the reviewer, we have compared the gene ranking for traits from different models. These new results (**New Figures S6 and S7, Lines 228-243**), show that the gene ranking is only highly correlated specifically with each trait (for instance, N uptake and biomass). This finding suggests that the gene feature importance ranking is specific to each trait.

4) As NUE is a cumulative polygenic trait, did the authors test alternative Maize RNA samples which were not N-responsive as a control? Can this approach work without a specific responsive sampling time/tissue to help predict unknown traits?

Response: To address this issue, we conducted new analyses in which we have compared our model results using the conserved N-responsive genes (N-DEGs), to randomly selected total expressed genes, or to top-ranked N-DEGs (with the lowest *P*-value), and these new results are shown in **New Table 1**. In both cases, the performance of the evolutionarily conserved N-DEGs is significantly better than models built from random expressed genes, or top-ranked N-DEGs. Both results support the importance of using an N-signaling/treatment and conservation which created a diverse panel of gene expression profiles.

5) An explanation on how this technique can be transferred to other systems and would be useful to back-up the last sentence of the abstract and in the discussion.

Response: To address this point, in the revised manuscript, we applied our pipeline to analyze a *rice* dataset published in a recent Nature paper (Groen *et al.*, 2020), which includes *matched* transcriptomic and phenotypic data from 220 rice genotypes subjected to drought treatment. These 220 rice genotypes consist of genetically divergent landraces and breeding lines that capture the genotypic and phenotypic diversity of domesticated rice. From this large dataset, we identified 57 rice lines that had no missing data – phenotypic or transcriptomic – to use in our analysis. From these 57 genotypes, we first identified drought-DEGs from 20 randomly selected genotypes as the features to predict the fecundity in the left-out 37 genotypes. We repeated the random selection 10 times and the mean Person's *r* was 0.62 (**New Figure S8 and Lines 244-256**). This new result not only provides a use case of our pipeline in a different data set for another species, but also showcases that transcript-based prediction can be conducted using a smaller sample size (20 genotypes in this case), as compared to GWAS and eQTL which require hundreds of samples (Korte and Farlow, 2013).

Reviewer #2 (Expertise: Machine learning in crop sciences):

The research work in this manuscript studies nitrogen use efficiency (NUE) in crop Maize by building a gene-to-trait model based on transcriptome analysis and a comparative study with Arabidopsis data. Experimental phenotypic and genomic data about NUE were collected and analyzed by a pipeline built with a collection of bioinformatics, statistical and machine learning tools. While the study generates valuable new phenotypic and genomic data for NUE phenotype and led to interesting findings of gene-to-trait models, there is a flaw in the design of data analysis pipeline and some important analyses are missing. The introduction section does not provide a useful literature review and related work is completely missing.

We thank Reviewer #2 for bringing up critical questions and constructive criticisms. We have conducted additional analyses included in the revised manuscript - that address each question raised by the reviewer, as detailed below.

Major points:

1. The machine learning analyses were intended for both predictive and descriptive modeling, which is a strength of this study. While the result of descriptive modeling for selecting the NUE driver genes, the evaluation of the predictive modeling is inaccurate and unacceptable (e.g. results in Table 1 and Step 3 in Figure 5). *In particular, The differential gene expression analysis with general linearized model already utilized phenotypic data (NUE output).* Evaluating how the selected genes are predictive of the NUE output in a machine learning model is an over-optimistic evaluation since the output of the test set in the leave-one-out-cross-validation has already been used to identify the genes. This leak of test label information is particularly problematic for evaluation on datasets of small size since some features could be selected to capture a particular sample in the small dataset.

Thus, the predictive modeling should be re-evaluated in a separate cross-validation. In particular, the cross-validation should be applied to the analysis in Fig 3 and Fig 5 together, i.e. a test sample is held out before the differential gene expressions analysis and the analysis should be repeated for every held-out sample. Without this rigorous evaluation, the predictive modeling is not validated as claimed.

The responses to this multi-part critique are below:

1.1 The evaluation was over-optimistic as the DEG analysis already utilized the NUE data.

1.1 We would like to clarify that our differential gene expression (DEG) analysis with general linearized model **did not** utilize the phenotypic data (NUE). Thus, there was no “leak” of test label information in our current or previous analysis: we only utilized gene expression data in the DEG analysis.

1.2 The cross-validation should be applied to the DEG analysis.

1.2 We have adopted Reviewer 2’s suggestion to use a “leave-out-one” strategy in our DEG analysis. To this end, we re-analyzed the data following this suggestion: a genotype was held out before the DEG analysis. We repeated this for each genotype and used the DEGs shared by all “leave-out-one” DEG analyses as the gene features in our XGBoost models. We found the performance of the predictive modeling remains comparable to our original one, even with this more rigorous evaluation. The revised pipeline - using the “leave-out-one” in the DEG analysis, and these new results are shown in **New Figures 4 and 5** and described in **Lines 152-156**.

2. While the choices of the methods such as generalized linear model and XGboost (random forests) seem to be reasonable, there is insufficient comparison with baselines or other commonly used alternative methods. In particular, the comparisons in Table 1 are not useful with randomly chosen genes or N-DEGs. How does the top genes selected by evolutionarily conserved N-DEGs compared with the same number of original top DEGs without using the evolutionary analysis? Other than generalized linear model, DEG analysis should also be compared with commonly used t-test and Wilcoxon ranksum test. In addition, how different is the ranking by XGboost compared with the original ranking by p-values in statistical analysis after filtering with evolutionary information? This comparison is necessary to justify that XGboost captures features beyond single variable analysis.

Responses to this multi-part comment are below.

2.1 How does the top genes selected by evolutionarily conserved N-DEGs compared with the same number of original top DEGs without using the evolutionary analysis?

2.1 As suggested by Reviewer 2, in the revised manuscript, we have conducted a comparison of model performance based on using three distinct sets of genes: Evolutionarily conserved N-DEGs, Top-ranked N-DEGs, and Random expressed genes. The outcome is that the models built from Evolutionarily conserved N-DEGs outperformed the ones build from the same number of N-DEGs with the lowest *P*-value in the DEG analysis, and also for Random expressed genes. This finding holds true for both Arabidopsis and maize. These new results are now shown in **New Table 1** and described in **Lines 198-221**.

2.2 Some comparison of different DEG analysis methods.

2.2 We agree with the Reviewer 2 that GLM is not the only method for DEG analysis. ANOVA, however, requires normal distribution, which is not the distribution of RNA-seq data. In addition, a simple t-test for RNA-seq can't accurately estimate variance unless a large sample size is available (Ritchie et al., 2015). Thus, we think advanced modeling is essential for RNA-Seq data analysis. One of the biggest advantages is the gene variance estimation was improved by pooling information from all genes (Robinson and Smyth, 2008). This method was first introduced in limma (Smyth, 2004) then adopted by edgeR (Robinson et al., 2010) and DESeq2 (Love et al., 2014). All three methods (limma, edegR and DESeq2) are widely used and show consistent and comparable good performance as demonstrated in several benchmark evaluations (Schurch et al., 2016; Costa-Silva et al., 2017; Baik et al., 2020).

2.3 How different is the ranking by XGboost compared with the original ranking by p-values?

2.3 To address this question, in the revised manuscript we have analyzed the relationship between XGBoost feature importance ranking and the edgeR *P*-value ranking (**New Figure S4**). In this new analysis, we have found the Spearman's rank correlation coefficient *rho* is 0.14 for Arabidopsis, and 0.19 for maize (**New Figure S4**). This new result supports our hypothesis that XGBoost models capture non-linear gene-trait relationships and is consistent with the new result described in further in Author Response **2.1**, that the models built from Evolutionarily conserved N-DEGs outperformed the ones using the same number of top N-DEGs selected from edgeR-based *P*-Value ranking (**New Figure S4, as discussed in Lines 198-221**).

3. The introduction section about machine learning and "curse-of-dimensionality" is too broad and irrelevant to this study given the audience of Nature Communication are from a broader community of many other research fields. There are many other machine learning methods and pipelines were proposed for comparative studies across species. For example, people use canonical correlation analysis (CCA) to co-project two different datasets. The contribution of this study is limited to plant species unless the authors include a comprehensive survey of (and a comparison with) those comparative

genomics methods such as those many methods for integrative analysis of human and mouse transcriptomes.

Below are our responses and revisions to this multi-part suggestion.

3.1 The introduction about "curse-of-dimensionality" is too broad and irrelevant to this study.

3.1 We believe "curse-of-dimensionality" is relevant to this study, because omics data, e.g. the transcriptome data we used in our study, by nature are highly dimensional. The number of features (gene loci, SNPs, epigenomic features, etc) inevitably outnumber the sample size, resulting in a common challenge in biological studies (Xu and Jackson, 2019). Our approach to use evolutionary conservation as a biologically principled feature reduction method, seeks to address this common "curse-of-dimensionality" – which fits the interest of the audience of Nature Communication coming from a diverse research field.

3.2 Comparison of additional comparative genomics method, e.g., CCA.

3.2 We acknowledge there are many other machine learning methods other than XGBoost. Indeed, CCA can identify linear combinations of features that have maximum correlation with each other. However, applying CCA to high-dimensional data remains challenging. First, when the gene set size much bigger than the sample size, the solution of the canonical variables are likely to be non-unique (Waaajenborg et al., 2008), in such cases the canonical correlations can be over-estimated (Pezeshki et al., 2004). Although there are methods, including penalized CCA (Witten et al., 2009), PCA-CCA (Song et al., 2016) and sparse CCA (Witten and Tibshirani, 2009), to remedy this issue, their effect on bulk RNA-Seq remains largely untested. Second, only one-to-one mapping orthologs can be used for CCA analysis. Only 16% (4798/29311) of the Arabidopsis – maize orthologues belong to one-to-one mapping orthologs. As such, we will be missing most of the genes if we use the CCA method. Lastly, the CCA method is unable to identify the non-linear combinations among variables.

3.3 The contribution of this study is limited to plant species unless the authors include a comprehensive survey of (and a comparison with) those comparative genomics methods such as those many methods for integrative analysis of human and mouse transcriptomes.

3.3 We believe that a main strength of our manuscript lies in the generation of *matched* transcriptomic and phenotypic data from *the same samples*, in both a model and a crop species. This matched experimental design allowed us to predict traits using a much smaller sample size (number of genotypes = 20), compared with GWAS and eQTL studies which typically require hundreds of samples (Korte and Farlow, 2013).

To prove the portability of our pipeline, we in a new analysis, we have successfully used additional a large set of rice gene expression and phenotype data sets (Groen *et al.*, 2020), as detailed in **Response 5 for Reviewer #1**. These new results are shown in **New Figure S8 and Lines 244-256**. To our surprise, we were unable to find animal data sets including matched gene expression and traits measured from the same samples.

Minor points:

1. The analysis in Fig. 2 is still confusing. What are the phenotypic features used in SVD? What's their scale (mean and variance)? It appears Nitrogen use is not a feature used in SVD analysis of Arabidopsis data, which should be clarified. The variance of each ET should also be provided. ET1-ET3 should also be visualized to show the relevance to the phenotypic features.

Response: We have chosen a different visualization to show that the primary explanatory factor for the variation of NUE traits is nitrogen and genotype in Arabidopsis and maize respectively, as in **New Figures 2c and 3c**.

2. In Figure 3, the superscript 1 and 2 above Feature Reduction is unannotated.

Response: We have fixed this annotation in the **New Figure 4**.

3. The heat map of DEG expressions in Figure 4 is not properly annotated and not helpful for the understanding of the research work. If there is not enough space to show the gene names, this figure should be moved to supplementary document, a figure showing a subset of most interesting genes can be included.

Response: We agree with the reviewer that the heatmap is not helpful in understanding the research work. Instead of showing a heatmap of all N-DEGs, we have now instead highlighted the N-DEGs with the lowest *P*-value in the DEG analysis, as requested by Reviewer 2 (Major Point 2). We have now compared and demonstrated that the models using conserved N-DEGs outperformed those using N-DEGs with the lowest *P*-value. These results are shown in **New Figure S4 and Table 1** and replace of the heatmap in the revised manuscript.

4. In addition, since there is no evidence or comparison with other similar data integration methods, the last statement (line 67-69) on page 2 is not well supported by this study.

Response: To prove the portability of our pipeline, we have successfully used additional large set of matched gene expression and phenotype data sets from rice (Groen *et al.*, 2020), as detailed in **Response 5 for Reviewer #1**. These new results are shown in **New Figure S8 and Lines 244-256**. To our surprise, we were unable to find animal data sets including *matched* gene expression and traits measured from the same samples.

5. The sentence between line 48-49 on page 2 seems not to be logical after the previous paragraph about "curse-of-dimensionality". Why exploiting transcriptome data to predict nitrogen use efficiency can address the afore mentioned computational problems?

Response: We have revised these sentences (**Lines 48-50**).

Reviewers' Comments:

Reviewer #1:

Remarks to the Author:

The revised manuscript of Cheng et al, is much improved and importantly, structurally more sound and informative to the wider readership of Nature Communications. Similar to the first read of the manuscript, I support the direction that this research has taken. The outcomes are exciting and I expect will have immediate impact in addressing a challenging genetic bottleneck in improving NUE or at least standardising the NUE performance properties of many crop plants that we rely on for daily dietary needs. New data, interpretation and expanded use of the genetic pipeline all together provides a convincing story and approach to use unbiased genetics to define or predict phenotypical outcomes.

The authors have carefully responded to my questions and in the revised manuscript, including new data, improved data presentation, and self-reflection in an unbiased manner that strengthens the data and message but importantly provides a tool which enriches the general NUE field. I appreciate the detail presented regarding the background of the plant populations and the significant backstory regarding how lines were selected from lab and field-based experiments. This is important as it helps to highlight one of the main outcomes of the study that carefully curated small plant populations with defined treatments and careful phenotypical evidence can be effective tools to predict gene to phenotype relationships, an issue which hampers other approaches, including eQTL. I wasn't able to see the Supplementary Table #1 in the review documents but I am assuming the raw data sets linked to the review, which I looked at, covers the majority of data to be presented in Supp Table 1.

I'm not surprised that additional traits can also be scoped using the gene sets identified. I appreciate including this analysis in Figure S5, it is a good addition to the manuscript. Similarly, the improved performance of the model with evolutionary conserved N-DEGS adds weight to the model and overall outcome of the experiments. Most importantly, the use of the pipeline model on a completely divergent gene expression data set, rice lines subject to drought, highlights the robustness of the approach and its broad application and utility across different topics. I appreciate the extra data (Fig S8) to show how the pipeline can be used for future or historical data sets.

In summary, the revised manuscript is much improved and I believe will become an important starting point to help elucidate polygenic trait descriptions through an evolutionary conserved genetic approach that removes experimental bias and pre-conceived directions.

Note:

A web-portal that would allow other researchers to input data and run the machine models would be a useful extension to this work?

Reviewer #2:

None

Reviewer #3:

Remarks to the Author:

While a couple of steps were taken to address the previous concerns, some of the major concerns are not completely addressed yet. I'll indicate this point by point, with the numbering scheme used in the rebuttal.

1.1 Here the authors argue that because NUE is not used, there is no information leakage between the feature selection stage (DEG analysis) and prediction stage (XGBoost model). It is good that it is clarified that NUE itself is not directly used. However, the description in the manuscript indicates that N is used (line 158, expression $\sim N + \text{genotype}$). Given that N is also an important factor in explaining variation in NUE, there clearly is a risk for information leakage, which should be addressed.

1.2 The information leakage issue should be addressed by a leave-one-out approach, but that should be implemented differently from what is described in the revised version of the manuscript. The approach described now does **not** leave-out phenotype-related information (about N) before the prediction of NUE is made - the leave-one-out results are currently all combined at the DEG stage before moving to the XGBoost stage. Instead, what should be done is that a given test sample in the cross-validation is held out before the DEG stage and then is not used at all during the DEG stage and subsequent XGBoost model training stage. A prediction can then be made for such test sample (which has not been used at all during DEG analysis or XGBoost training) to get an unbiased test performance estimation. The original comment that such rigorous evaluation is needed, has not been addressed by the current leave-one-out at the DEG stage.

2. Comments on comparison with baseline methods - this has been sufficiently addressed.

3.1 I am not completely convinced by the response. I agree that the curse of dimensionality is a relevant point, but the introduction is rather broad, and could be more focused on using comparative genomics approaches in the context of ML.

3.2 The response contains some valid issues but this does not directly mean that an alternative method such as CCA could not be mentioned in the introduction. Note by the way that for some of the concerns that are mentioned w.r.t. to CCA by the authors, I am wondering if these are also not concerns w.r.t. their own study (influence of orthology detection, and general problem that gene set is larger than sample set).

3.3 Inclusion of an extra analysis is fine and addresses this point to some level, but in terms of introducing the idea of cross-species application of ML to predict phenotypes, this does not help much. As suggested previously, many methods have been suggested in animals, in particular related to the combination mouse-human. One typical example is e.g.

<https://journals.plos.org/ploscompbiol/article?id=10.1371/journal.pcbi.1006286>. The remark that animal sets with matched gene expression and traits are not available is not correct. A large example is e.g. given by the GTEx data which focuses on transcriptome data but also includes e.g. height, bmi, and various descriptors related to medical conditions.

As for the minor concerns, these are mostly addressed in a satisfactory way, except those related to the introduction (dealt with above in 3).

REVIEWER COMMENTS

Reviewer #1 (Remarks to the Author):

The revised manuscript of Cheng et al, is much improved and importantly, structurally more sound and informative to the wider readership of Nature Communications. Similar to the first read of the manuscript, I support the direction that this research has taken. The outcomes are exciting and I expect will have immediate impact in addressing a challenging genetic bottleneck in improving NUE or at least standardising the NUE performance properties of many crop plants that we rely on for daily dietary needs. New data, interpretation and expanded use of the genetic pipeline all together provides a convincing story and approach to use unbiased genetics to define or predict phenotypical outcomes.

The authors have carefully responded to my questions and in the revised manuscript, including new data, improved data presentation, and self-reflection in an unbiased manner that strengthens the data and message but importantly provides a tool which enriches the general NUE field. I appreciate the detail presented regarding the background of the plant populations and the significant backstory regarding how lines were selected from lab and field-based experiments. This is important as it helps to highlight one of the main outcomes of the study that carefully curated small plant populations with defined treatments and careful phenotypical evidence can be effective tools to predict gene to phenotype relationships, an issue which hampers other approaches, including eQTL. I wasn't able to see the Supplementary Table #1 in the review documents but I am assuming the raw data sets linked to the review, which I looked at, covers the majority of data to be presented in Supp Table 1.

I'm not surprised that additional traits can also be scoped using the gene sets identified. I appreciate including this analysis in Figure S5, it is a good addition to the manuscript. Similarly, the improved performance of the model with evolutionary conserved N-DEGS adds weight to the model and overall outcome of the experiments. Most importantly, the use of the pipeline model on a completely divergent gene expression data set, rice lines subject to drought, highlights the robustness of the approach and its broad application and utility across different topics. I appreciate the extra data (Fig S8) to show how the pipeline can be used for future or historical data sets.

In summary, the revised manuscript is much improved and I believe will become an important starting point to help elucidate polygenic trait descriptions through an evolutionary conserved genetic approach that removes experimental bias and pre-conceived directions.

Note:

A web-portal that would allow other researchers to input data and run the machine models would be a useful extension to this work?

We acknowledge the positive comments from the Reviewer 1. While we are unable to construct a web-portal at the moment, we have deposited all the codes and input files to the OSF depository, which will allow researchers to build a prediction models using their own data.

Reviewer #3 (Expertise: machine learning with an application to crop science):

While a couple of steps were taken to address the previous concerns, some of the major concerns are not completely addressed yet. I'll indicate this point by point, with the numbering scheme used in the rebuttal..

1.1 Here the authors argue that because NUE is not used, there is no information leakage between the feature selection stage (DEG analysis) and prediction stage (XGBoost model). It is good that it is clarified that NUE itself is not directly used. However, the description in the manuscript indicates that N is used (line 158, expression $\sim N + \text{genotype}$). Given that N is also an important factor in explaining variation in NUE, there clearly is a risk for information leakage, which should be addressed.

This point is about “biological” information leakage, namely that N-treatments affect both N-DEG and NUE. First, we addressed this issue by comparing the models generated using top-ranked N-DEGs vs. those using evolutionarily conserved N-DEGs for their ability to predict NUE (Table 1). This analysis shows that the models generated by the N-response DEGs conserved across Arabidopsis and maize outperform the ones generated by top-ranked N-DEGs. Second, our comparison of the feature importance score, and XGBoost output (which reveals the influence of each gene feature on the predicted value (NUE)), with the *P*-value in DEG analysis, uncovered only a weak correlation (Supplementary Fig. 5). The two results each support that the N-DEG responsiveness alone is insufficient to explain the learned model performance in predicting the NUE traits, compared to the evolutionarily conserved N-DEGs, which is a main point of our study.

1.2 The information leakage issue should be addressed by a leave-one-out approach, but that should be implemented differently from what is described in the revised version of the manuscript. The approach described now does *not* leave-out phenotype-related information (about N) before the prediction of NUE is made - the leave-one-out results are currently all combined at the DEG stage before moving to the XGBoost stage. Instead, what should be done is that a given test sample in the cross-validation is held out before the DEG stage and then is not used at all during the DEG stage and subsequent XGBoost model training stage. A prediction can then be made for such test sample (which has not been used at all during DEG analysis or XGBoost training) to get an unbiased test performance estimation. The original comment that such rigorous evaluation is needed, has not been addressed by the current leave-one-out at the DEG stage.

This comment is about information leakage in machine learning, where information outside the training set data is used to create the model. We agree that this is an important point, and note that our DEG analysis in the manuscript avoids this issue, as it is essentially the same as the reviewer's suggestion. However, we agree that our explanation of this point was not clear, and we have revised the text to clarify this issue (Lines 159-163): *“we used negative binomial Generalized Linear Mixed models (GLMs) in edgeR R-package and identified N-DEGs (Gene expression \sim Condition + Genotype) in the training data (n-1 genotype). Importantly, we note that the testing data (the held-out genotype) was never used to select the N-DEGs. This was repeated in a round-robin manner for each species.”* To stress and clarify this point, we have added a new Supplementary Figure 4 to illustrate how the feature selection was done for DEG (see below).

New Supplementary Figure 4. Comparison of XGBoost models created using a unified list of gene features (a), or independent lists of gene features (b). **(a)** Unified list of gene features: For each round of DEG analysis, a test genotype was held-out and not included in the DEG analysis. This resulted in 18 independent DEG lists for Arabidopsis (a, i). These 18 independent DEG lists were then intersected to generate a unified list of DEGs within species (a, ii). Next, the DEGs shared by all 18 Arabidopsis genotypes were used for cross-species intersection with maize (a, iii). This approach resulted in a unified list of gene features which was used in XGBoost models (a, iv). **(b)** Independent list of gene features: This approach used 18 independent DEG lists within Arabidopsis (b, i), and each list was intersected independently in a cross-species analysis with maize (b, iii), and used in XGBoost models in (b, iv). (Note: The numbers of gene features shown in Panels a and b are for Arabidopsis. However, the 16 maize samples were processed using this same pipeline). **(c)** Model performance: The XGBoost performance using a unified list of DEGs (a), is comparable to the independent DEG list (b), for both Arabidopsis and maize. The advantage of using the unified list approach (a), as done in this study, is that it generates a ranked list of gene feature importance that is relevant across many genotypes.

Indeed, as Reviewer 3 suggested, we held out a testing genotype **before** the DEG stage; and only training genotypes ($n-1$ genotypes) were used in DEG analysis and XGBoost models (as depicted in New Figure S4). The held-out test genotype was then used to validate the model performance. This round robin approach (New Fig. S4a(i) & b(i)), generated 18 and 16

independent DEG lists for Arabidopsis and maize, respectively. In approach a, we identified a unified list of gene features by intersecting these independent lists (e.g. 18 for Arabidopsis and 16 for maize) (Fig. S4a(ii)). By contrast, in approach b, cross species analysis was performed on each independent DEG list (e.g. 18 for Arabidopsis or 16 for maize).

Moreover, to rule out the possibility that using the intersected DEGs (e.g. within species) would overly optimize the XGBoost results, we further compared the XGBoost performance using the intersected DEGs (Fig. S4a) with the alternative approach that did not go through the within species list intersection (Fig. S4b). The results of these two approaches are comparable (Fig. S4c). However, the advantage of conducting the cross-genotype intersection (Fig. S4a, which we used in this manuscript), has the benefit of resulting in a unified list of gene features, compared to multiple independent lists of gene features. Generating a unified list of gene features will enable the gene feature ranking across genotypes, rather than restricted to an individual genotype. We also included these clarifications in Methods (Lines 579-593).

2. Comments on comparison with baseline methods - this has been sufficiently addressed.

3.1 I am not completely convinced by the response. I agree that the curse of dimensionality is a relevant point, but the introduction is rather broad, and could be more focused on using comparative genomics approaches in the context of ML.

3.2 The response contains some valid issues but this does not directly mean that an alternative method such as CCA could not be mentioned in the introduction. Note by the way that for some of the concerns that are mentioned w.r.t. to CCA by the authors, I am wondering if these are also not concerns w.r.t. their own study (influence of orthology detection, and general problem that gene set is larger than sample set).

To acknowledge this input from Reviewer 3, we have removed the term “curse of dimensionality” and mentioned additional commonly used feature reduction methods in the revised manuscript. In the revised text, we have also further emphasized the strength of integrating genetic diversity and cross-species as dimension reduction approaches in machine learning (Lines 39-46): *“Furthermore, the explosion of omics data means that the features (e.g. numbers of genes) collected from a single experiment inevitably outnumber the phenotype space (e.g. sample size), leading to problems in data sparsity, multicollinearity, multiple testing, and overfitting³. This can be counteracted with increasing sample size, dimension reduction or feature selection methods such as Principal Component Analysis (PCA), Least Absolute Shrinkage and Selection Operator (LASSO) regularization, Canonical Correlation Analysis (CCA), and so forth⁴. Additionally, the cross-species approach has been adopted in machine learning contexts to improve the performance of model-to-human knowledge translation⁵.”*

3.3 Inclusion of an extra analysis is fine and addresses this point to some level, but in terms of introducing the idea of cross-species application of ML to predict phenotypes, this does not help much. As suggested previously, many methods have been suggested in animals, in particular related to the combination mouse-human. One typical example is e.g. <https://journals.plos.org/ploscompbiol/article?id=10.1371/journal.pcbi.1006286>. The remark that animal sets with matched gene expression and traits are not available is not

correct. A large example is e.g. given by the GTEx data which focuses on transcriptome data but also includes e.g. height, bmi, and various descriptors related to medical conditions.

In our revised manuscript, we conducted analyses to apply our evolutionarily informed machine learning pipeline to two external data sets from diverse genotypes of a plant (rice) and an animal species (mouse). These new results (below), conclusively support a main finding in this manuscript: that conserved DEGs - either cross-species or -subspecies with divergent genetic backgrounds - enhance the predictive power of gene-to-trait relationships, as summarized below:

Rice: Regarding our application of our approach to another species (rice), we note that genotypes used in the rice study include two major **subspecies**, Indica and Japonica, which diverged ~440,000 years ago (Garris et al., 2005). Thus, we were able to provide this use-case of our pipeline to an external data set and show that evolutionarily conserved DEG (e.g. across rice subspecies) can accurately predict a complex trait across genotypes.

Mouse: We agree with Reviewer 3 that our previous remark about the available cross-species animal data was inaccurate. To address this point, in the revised manuscript, we have performed additional analyses using our pipeline to exploit genetic diversity in animals. To this end, we sought a comparable transcriptome/phenotype in animals to test our pipeline. As there is a growing awareness in biomedical research that host genetic diversity has a major impact on complex traits (Sirugo et al., *Cell*, 2019), we believe studies using Collaborative Cross (CC) mouse lines would be of suitable genetic variability for our approach. The CC lines represents a mouse recombinant inbred strain panel that comprises 90% of the genetic diversity across the entire laboratory *Mus musculus* genome. We searched on GEO and found 49 studies that utilized CC mice in their work. We carefully reviewed each of the studies and found that phenotypes were only measured in 10 of them. To our surprise, we only found one study (Kollmus et al., 2018) that made publicly available both transcriptome and phenotype on 11 genotypes of CC mice infected with influenza virus. We thus re-analyzed the CC mouse data using our evolutionarily informed machine learning pipeline and now include these results in the revised manuscript (Lines 264-267): "*We used DEG (mock vs. infected) across 11 mouse CC population genotypes to predict the disease outcome (asymptomatic vs. symptomatic) and found the mean Pearson's r to be 0.98. The models built using cross-genotype DEGs from these genotypes outperformed the model using the same number of random expressed genes (Mann–Whitney U test, P-value = 3.3E-3).*" We did not find comparable studies in human, as the existing studies (Woods et al., 2013, Zhai et al., 2015, Tang et al., 2017) only include transcriptome data.

As for the minor concerns, these are mostly addressed in a satisfactory way, except those related to the introduction (dealt with above in 3).

Reviewers' Comments:

Reviewer #3:

Remarks to the Author:

In the revision of their manuscript, Cheng et al address the issues which were raised before. It is good to see issues that were still unclear now clarified. The method presented here, and the outcomes on specific datasets, are relevant for a broad audience interested in analyzing the connection between genotype and phenotype. Cheng et al carefully responded to my comments and included additional datasets, and an improved description of their methodology. My previous doubts on aspects of the performance assessment have been dealt by these additions in a satisfactory way.

The revision strengthens the message of Cheng et al and will help to position this paper as an important source for readers interested in questions related to using machine learning and data analysis across species.